

# Short and long-term effects of a synbiotic on clinical signs, the fecal microbiome, and metabolomic profiles in healthy research cats receiving clindamycin: a randomized, controlled trial

Jacqueline C. Whittemore[1], Jennifer E. Stokes[1], Nicole L. Laia[1], Joshua M. Price[2] and Jan S. Suchodolski[3]

[1] Department of Small Animal Clinical Sciences, University of Tennessee—Knoxville, Knoxville, TN, United States of America
[2] Office of Information Technology, University of Tennessee—Knoxville, Knoxville, TN, United States of America
[3] The Gastrointestinal Laboratory, Small Animal Clinical Sciences, Texas A&M University, College Station, TX, United States of America

Corresponding author
Jacqueline C. Whittemore,
jwhittemore@utk.edu

## ABSTRACT

**Background**. Antibiotic-associated gastrointestinal signs (AAGS) occur commonly in cats. Co-administration of synbiotics is associated with decreased AAGS in people, potentially due to stabilization of the fecal microbiome and metabolome. The purpose of this double-blinded randomized-controlled trial was to compare AAGS and the fecal microbiome and metabolome between healthy cats that received clindamycin with a placebo or synbiotic.

**Methods**. 16 healthy domestic shorthair cats from a research colony were randomized to receive 150 mg clindamycin with either a placebo (eight cats) or commercially-available synbiotic (eight cats) once daily for 21 days with reevaluation 603 days thereafter. All cats ate the same diet. Food consumption, vomiting, and fecal score were recorded. Fecal samples were collected daily on the last three days of baseline (days 5–7), treatment (26–28), and recovery (631–633). Sequencing of 16S rRNA genes and gas chromatography time-of-flight mass spectrometry was performed. Clinical signs, alpha and beta diversity metrics, dysbiosis indices, proportions of bacteria groups, and metabolite profiles were compared between treatment groups using repeated measures ANOVAs. Fecal metabolite pathway analysis was performed. $P < 0.05$ was considered significant. The Benjamini & Hochberg's False Discovery Rate was used to adjust for multiple comparisons.

**Results**. Median age was six and five years, respectively, for cats in the placebo and synbiotic groups. Hyporexia, vomiting, diarrhea, or some combination therein were induced in all cats. Though vomiting was less in cats receiving a synbiotic, the difference was not statistically significant. Bacterial diversity decreased significantly on days 26–28 in both treatment groups. Decreases in *Actinobacteria* (*Bifidobacterium, Collinsella, Slackia*), *Bacteriodetes* (*Bacteroides*), *Lachnospiraceae* (*Blautia, Coprococcus, Roseburia*), *Ruminococcaceae* (*Faecilobacterium, Ruminococcus*), and *Erysipelotrichaceae* (*Bulleidia, [Eubacterium]*) and increases in *Clostridiaceae* (*Clostridium*) and *Proteobacteria* (*Aeromonadales, Enterobacteriaceae*) occurred in both treatment groups, with

incomplete normalization by days 631–633. Derangements in short-chain fatty acid, bile acid, indole, sphingolipid, benzoic acid, cinnaminic acid, and polyamine profiles also occurred, some of which persisted through the terminal sampling timepoint and differed between treatment groups.

**Discussion**. Cats administered clindamycin commonly develop AAGS, as well as short- and long-term dysbiosis and alterations in fecal metabolites. Despite a lack of differences in clinical signs between treatment groups, significant differences in their fecal metabolomic profiles were identified. Further investigation is warranted to determine whether antibiotic-induced dysbiosis is associated with an increased risk of future AAGS or metabolic diseases in cats and whether synbiotic administration ameliorates this risk.

# INTRODUCTION

Antibiotic administration is associated with profound, and sometimes prolonged, derangements of the fecal microbiome and metabolome of people and animals (*De La Cochetière et al., 2010*; *Dethlefsen et al., 2008*; *Jakobsson et al., 2010*; *Suchodolski, 2016b*; *Suchodolski et al., 2009*). This dysbiosis is believed to be a primary contributor to the development of antibiotic-associated gastrointestinal signs (AAGS), such as hyporexia, vomiting, and diarrhea (*Hempel et al., 2012*; *McFarland, 2008*; *Videlock & Cremonini, 2012*). Susceptibility to AAGS and derangements in the microbiome both increase with repeated antibiotic exposure (*Ouwehand et al., 2016*).

Antibiotic-associated gastrointestinal signs occur commonly in cats (*Albarellos & Landoni, 2009*; Albarellos & Landoni, 1995; *Hunter et al., 1995*) and people (*Hempel et al., 2012*; *Lenoir-Wijnkoop et al., 2014*; *McFarland, 2008*), and they are an important cause of antibiotic non-compliance (*Chan et al., 2012*; *Jefferds et al., 2002*; *Kardas et al., 2005*; *Llor et al., 2013*; *Muñoz et al., 2014*; *Pechere et al., 2007*). Co-administration of probiotics or synbiotics (commercial mixtures of probiotics and prebiotics) with antibiotics is associated with up to a 3-fold decrease in AAGS in people (*Hempel et al., 2012*; *Johnston et al., 2011*; *Lenoir-Wijnkoop et al., 2014*; *Narayan et al., 2010*; *Selinger et al., 2013*). Positive effects of probiotics on AAGS in people are postulated to result from stabilization of the fecal microbiome and metabolome (*Hempel et al., 2012*; *Ouwehand et al., 2016*).

The impact of synbiotics on AAGS and antibiotic-induced dysbiosis in cats is currently unclear. However, the administration of one commercially-available multi-strain synbiotic (Proviable-DC; Nutramax Laboratories Veterinary Sciences, Inc., Lancaster, SC, USA) resulted in significantly improved fecal scores for 72% of cats with idiopathic chronic diarrhea (*Hart et al., 2012*), suggesting potential efficacy for reducing gastrointestinal signs due to other causes. Prevention or mitigation of AAGS could decrease noncompliance by

clients and, thus, patient morbidity. Even if direct clinical effects are not observed, synbiotic administration might be warranted if it is found to decrease antibiotic-induced changes in the microbiome and metabolome and, thus, potentially reduce cumulative long-term sensitivity to AAGS and/or metabolic disorders associated with antibiotic exposure, such as inflammatory bowel disease and obesity (*Aniwan et al., 2018*; *Kronman et al., 2012*; *Ouwehand et al., 2016*).

The purpose of this study was to compare changes in food intake, vomiting, fecal scores, the fecal microbiome, and the fecal metabolome of healthy cats receiving either a placebo or a commercially-available synbiotic with 150 mg clindamycin orally once daily for 21 days. A secondary objective of the study was to assess the long-term impacts of antibiotic and synbiotic administration on the fecal microbiome and untargeted metabolomic profiles of healthy cats.

## MATERIALS AND METHODS

### Study population

This study was performed at the University of Tennessee's Veterinary Medical Center and was approved by the Institutional Animal Care and Use Committee at the University of Tennessee, Knoxville (protocol number 2169). Twenty purpose-bred, domestic short hair cats from the University of Tennessee, College of Veterinary Medicine teaching and research colony determined to be healthy based on unremarkable physical examinations at the time of study initiation and lack of pertinent abnormalities on CBC or biochemistry profiles performed within the previous six months were enrolled. Prior antibiotic use was not noted in any of the medical records. All cats ate the same commercial adult dry cat food for the duration of the study (Hill's Science Diet Light; Hill's Pet Nutrition, Inc., Topeka, KS, USA). Cats were transitioned to individual housing the week prior to the start of the study and randomized into 1 of 2 treatment groups (placebo vs. synbiotic) at the start of the study using a random number sequence.

### Study periods

The study consisted of a baseline period (days 1–7), a treatment period (days 8–28), and a long-term follow-up period (days 631–633) (Fig. 1). During the baseline period, no treatments were administered. During the treatment period, all cats received 150 mg (34 mg/kg/d, range 25.9–44.1 mg/kg/d) of clindamycin (Antirobe Antirobe Aquadrops; Pfizer, Pharmacia & Upjohn Company, New York, NY, USA) PO once daily with food for 21 days. The dose was chosen based on the manufacturer drug insert (Antirobe Antirobe Aquadrops manufacturer insert, Pfizer Pharmacia, March 2010), which noted changes in fecal consistency and emesis in cats receiving higher doses of clindamycin. Cats received either one capsule of placebo or a commercially-available synbiotic PO once daily at the time of antibiotic administration. The synbiotic contains five billion *cfu* of a proprietary mixture of *Bifidobacterium bifidum, Enterococcus faecium* and *thermophilus,* and *Lactobacillus acidophilus, bulgaricus, casei,* and *lantarum* per capsule, as well a proprietary blend of fructooligosaccharide and arabinogalactan. The time period for treatment was chosen based on previous veterinary publications (*Garcia-Mazcorro et al., 2011*; *Hart et*

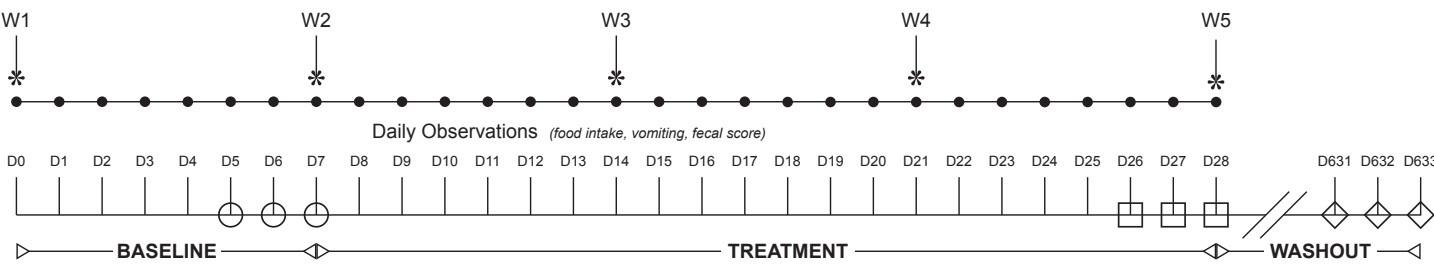

**Figure 1** **Flowchart illustrating study design, duration, observations, and sampling.** The study spanned 633 days (D1-633) and was broken into three study periods: baseline (D1–D7), treatment (D8–D28), and washout. Cats were randomized to receive either a placebo or synbiotic with 150 mg clindamycin PO once daily during treatment. Food intake, vomiting, and fecal score were recorded daily (●) and weight (W) weekly (*) by an individual blinded to treatment group. Two grams were collected from the center portion of each first morning naturally-voided fecal samples for each cat once daily for the last three days of each study period: baseline (open circles), treatment (open squares), and recovery (open diamonds). Each sample was subdivided into two aliquots, with each aliquot placed into a 2 mL cryovial and immediately frozen at −80 °C pending completion of data collection.

*al., 2012*; *Koeppel et al., 2006*), which showed significant changes in the fecal microbiome or fecal score by 21 days of treatment. After completion of treatment, cats were returned to group housing in the colony. On day 631, cats remaining in the teaching and research pool were reevaluated. Based on review of their medical records, none of the cats had received antibiotics, probiotics, other medications known to affect the microbiome, or dietary change in the intervening time.

## Data collection

All cats underwent weekly physical examinations, including determination of body weight, throughout the study. Daily observations, including food intake and vomiting (present vs. absent), were collected throughout the study by an observer (NLL) blinded to the treatment groups. Photographs of defecated feces were taken daily by the same observer. Fecal consistency was scored (*Greco, 2011*) using still photographs at the completion of data collection by two investigators (JCW and JES), who were blinded to the cat, treatment group, and day for each fecal sample.

## Fecal samples

To limit the impact of daily variation in bacterial populations and fecal metabolites, as well as differential distribution of bacterial groups and metabolites within individual fecal samples, first morning naturally-voided fecal samples were collected daily for three days for each cat at each of three timepoints: the conclusion of baseline (days 5–7), the conclusion of treatment (days 26–28), and after long-term recovery (days 631–633). Two grams from the center portion of each fecal sample was subdivided into two aliquots, with each aliquot placed into a 2 mL cryovial. Samples were immediately frozen and remained in storage at −80 °C pending completion of data collection. Samples for each cat from each timepoint were combined directly prior to sample analysis to generate pooled samples for microbiome and metabolomic analysis.

## Microbiome analysis

Genomic DNA was extracted from 100 mg of feces from each pooled sample using a commercially available kit according to manufacturer's protocol (PowerSoil; Mo Bio, Carlsbad, CA, USA).

Amplification and sequencing of the V4 variable region (primers 515F/806R) of the 16S rRNA gene was performed on a MiSeq (Illumina; MR DNA (Molecular Research LP), Shallowater, TX, USA) as previously described (*Bell et al., 2014*). The software Quantitative Insights Into Microbial Ecology (QIIME, v. 1.8; http://www.qiime.org) was used for processing and analyzing the sequences. The raw sequence data were de-multiplexed, and low quality reads were filtered using default parameters for QIIME. Chimeric sequences were filtered from the reads using USEARCH against the 97% clustered representative sequences from the Greengenes database (v. 13.8) and removed. The remaining sequences were assigned to operational taxonomic units (OTUs) using an open-reference OTU picking protocol using the default QIIME parameters, uclust consensus taxonomy assigner, and Greengenes database. The OTU table was rarefied to 35,000 sequences per sample. Alpha rarefaction plots, alpha diversity metrics (Chao1, Shannon, Goods Coverage, and Observed Species), and beta diversity metrics (weighted and unweighted UniFrac distance matrices) were created using QIIME scripts.

Quantitative PCR was performed for selected bacterial groups (total bacterial, *Faecalibacterium* spp.*, Turicibacter* spp.*, Streptococcus* spp.*, Escherichia coli, Blautia* spp.*, Fusobacterium* spp.*, Clostridium hiranonis*) using extracted DNA as has been previously described (*AlShawaqfeh et al., 2017*). Briefly, 2 µl of normalized DNA (final concentration: 5 ng/ µl) was combined with 5 µl of a DNA-binding dye (SsoFast EvaGreen supermix; Bio-Rad Laboratories, CA, USA), 0.4 µl each of a forward and reverse primer (final concentration: 400 nM), and 2.6 µl of PCR water to achieve a total reaction volume of 10 µl. Oligonucleotide primers and probes, as well as respective annealing temperatures, are summarized in Table 1. Data were expressed as log amount of DNA (fg) for each particular bacterial group per 10 ng of isolated total DNA.

## Metabolomics analysis

10 mg of lyophilized feces from each pooled sample were analyzed at a metabolomics facility using gas chromatography time-of-flight mass spectrometry (GC-TOF-MS) and standardized protocols as described in detail elsewhere (*Fiehn et al., 2008*; *Minamoto et al., 2015*). Raw data files were processed using ChromaTOF v. 2.32. BinBase algorithm was applied to match spectra to database compounds (West Coast Metabolomics Core, University of California, Davis, CA, USA) or to characterize as an unknown compound as reported previously (*Honneffer et al., 2017*). Quantification was reported by peak height of an ion at the specific retention index characteristic of the compound across all samples.

## Statistical analyses

Descriptive statistics were generated for each parameter. Samples were analyzed for normality using the Shapiro–Wilk test and, for the presence of outliers, using box-and-whisker plots. Age and weight for the two treatment groups were compared using an
**Table 1 Target bacteria, oligonucleotide primers/probes, and annealing temperatures used for quantitative PCR analysis.**

| Target | Primer type | Sequence (5′-3′) | Annealing (°C) |
|---|---|---|---|
| Universal bacteria | Forward | CCTACGGGAGGCAGCAGT | 59 |
| | Reverse | ATTACCGCGGCTGCTGG | |
| *Blautia* spp. | Forward | TCTGATGTGAAAGGCTGGGGCTTA | 56 |
| | Reverse | GGCTTAGCCACCCGACACCTA | |
| *Clostridium hiranonis* | Forward | AGTAAGCTCCTGATACTGTCT | 50 |
| | Reverse | AGGGAAAGAGGAGATTAGTCC | |
| *Escherichia coli* | Forward | GTTAATACCTTTGCTCATTGA | 55 |
| | Reverse | ACCAGGGTATCTAATCCTGTT | |
| *Faecalibacterium* spp. | Forward | GAAGGCGGCCTACTGGGCAC | 60 |
| | Reverse | GTGCAGGCGAGTTGCAGCCT | |
| *Fusobacteria* spp. | Forward | KGGGCTCAACMCMGTATTGCGT | 51 |
| | Reverse | TCGCGTTAGCTTGGGCGCTG | |
| *Streptococcus* spp. | Forward | TTATTTGAAAGGGGCAATTGCT | 54 |
| | Reverse | GTGAACTTTCCACTCTCACAC | |
| *Turicibacter* spp. | Forward | CAGACGGGGACAACGATTGGA | 63 |
| | Reverse | TACGCATCGTCGCCTTGGTA | |

independent two-sample Student's $t$-test. Mean percent food intake, percent days of vomiting, and mean fecal scores were determined for each week within each study period (baseline and treatment weeks 1, 2, and 3). Mean food intake for each week in each study period was calculated as a percentage of food intake of each cat during the week prior to the start of the study. Inter-rater correlation coefficients were calculated for fecal scores. The mean of fecal scores assigned by the two investigators were used for all further statistical analyses. Mean food intake, percent days vomiting per week, and mean fecal score were compared between treatment groups using a multivariate repeated measures ANOVA. Treatment group (placebo or synbiotic), week, and cat were included as categorical variables. Treatment, week, and the treatment-by-week interaction were included as fixed effects. Week was included as a repeated measure with subject as cat. Age, sex, and weight were initially included as covariates but were not retained in the final models. Model assumptions regarding normally distributed residuals were verified with the Shapiro–Wilk test for normality. Model assumptions regarding equality of variances were verified with Levene's Test for Equality of Variances.

Global changes in microbiota communities (beta diversity) between individuals were determined using unweighted Unifrac distance metrics; principal coordinates analysis (PCoA) plots and rarefaction curves were plotted using QIIME software. The ANOSIM function in PRIMER 6 (PRIMER-E Ltd, Ivybridge, UK) was used to compare beta diversity metrics across time and between treatment groups (*Suchodolski et al., 2012*). A dysbiosis index was calculated through methods previously described (*AlShawaqfeh et al., 2017*). The dysbiosis index is a quantitative PCR-based assay that summarizes the quantitative abundances of 8 major bacterial groups in feces in 1 number. The higher the dysbiosis index, the more deviation of the microbiota from normobiosis. Global changes in untargeted

metabolomic profiles were determined using principal component analysis (PCA) plots developed using an online metabolomics software analysis suite (MetaboAnalyst 3.0: http://www.metaboanalyst.ca) as per standard protocols (*Xia et al., 2015*; *Xia & Wishart, 2011*). Pathway analysis was performed in the same software suite using the *Homo sapiens* pathway library, interquantile range data filtering, log transformation, and Pareto scaling.

Shannon indices, goods coverage, the Chao 1 metric, the dysbiosis index, proportions of bacteria groups, and fecal metabolite profiles were compared between treatment groups using a mixed model, split-plot repeated measures ANOVA that included fixed effects of treatment (placebo or synbiotic), time period, and treatment-by-time period interaction. Cat nested within group was included as a random effect. Age and sex were included as covariates in the initial analysis, but they were not retained in the final model due to lack of significant effects. The repeated measure of time period was accounted for in a repeated statement. A compound symmetry variance/covariance structure was incorporated into each model to account for the potential inclusion of constant covariates over time. The Shapiro–Wilk test of normality of the residuals was evaluated for each marker to confirm the assumption of normally distributed residuals had been met. Model assumptions regarding equality of variances were verified with Levene's Test for Equality of Variances. Differences in least squares means were determined for markers with significant main effect or interaction terms. Due to a marked lack of variability in microbiome results for cats on days 26–28 (see Results below), rank transformation had to be employed to allow convergence of the model for analysis of the proportions of bacteria groups and ensure the statistical assumptions regarding normally distributed residuals and equality of variances were met. Only bacteria taxa that were present in at least 50% of cats in $\geq 1$ group at $\geq 1$ time point were included in statistical analyses.

$P < 0.05$ was considered statistically significant. *P*-values were corrected for multiple comparisons on each phylogenetic level and for untargeted metabolomics using the Benjamini & Hochberg's False Discovery Rate (fdr). Publicly-accessible software packages (http://www.qiime.org; http://www.metaboanalyst.ca; SAS 9.4 release TS1M3; MedCalc 15.8: MedCalc, Ostend, Belgium; SAS 9.4 release TS1M3: SAS Institute Inc., Cary, NC, USA) were used for all analyses.

# RESULTS

## Study population

There were four female spayed and 6 male castrated cats initially enrolled in the placebo group and five female spayed cats and five male castrated cats in the synbiotic group. Median age was six years (range 3–8 years) for cats in the placebo group and five years (range 3–8 years) for cats in the synbiotic group. Median weight was 4.2 kg (range 3.4–5.4 kg) for cats in the placebo group and 4.3 kg (3.4–5.8 kg) for cats in the synbiotic group. No cat vomited any of the administered capsules during the study. Four cats were removed from the study for marked hyporexia (two placebo, one synbiotic) or high baseline fecal scoring (1 synbiotic).

After completion of treatment, cats were returned to the colony. On day 631, 15/16 of the cats that completed the trial were available for re-examination. One cat (from the

**Table 2 Antibiotic-associated gastrointestinal signs for cats that received clindamycin and a placebo or synbiotic.** Mean (± standard deviation) percent food intake compared to the week prior to initiation of the study, percent days vomiting per week, and mean fecal score for 16 healthy cats, eight per group, that received 150 mg clindamycin with either a placebo or synbiotic PO once daily for 21 days.

| | Baseline | Treatment | | |
| --- | --- | --- | --- | --- |
| | | Week 1 | Week 2 | Week 3 |
| **Food intake (%)** | | | | |
| − Placebo | 97.8 ± 4.1 | 89.9 ± 5.5 | 89.7 ± 11.3 | 88.1 ± 13.2 |
| − Synbiotic | 98.3 ± 4.1 | 92.7 ± 8.2 | 94.1 ± 8.4 | 92.7 ± 11.1 |
| **Vomiting (%)** | | | | |
| − Placebo | 0 ± 0 | 26.8 ± 22.2 | 14.3 ± 20.2 | 9.0 ± 17.0 |
| − Synbiotic | 1.8 ± 5.1 | 21.5 ± 15.3 | 9.0 ± 15.2 | 1.8 ± 5.1 |
| **Fecal score** | | | | |
| − Placebo | 2.6 ± 0.8 | 3.2 ± 0.8 | 3.5 ± 0.8 | 3.7 ± 1.3 |
| − Synbiotic | 2.7 ± 0.8 | 3.7 ± 0.9 | 3.8 ± 0.7 | 4.2 ± 0.4 |

synbiotic group) had been retired from the colony with subsequent private adoption. None of the cats had received antibiotics, probiotics, or other medications known to affect the microbiome in the intervening time, and none had undergone diet change. None of the cats were experiencing vomiting, abnormal fecal scores, or inadequate body condition scores.

## Antibiotic-associated gastrointestinal signs

Food intake, percent days vomiting, and fecal scores over time by treatment group are summarized in Table 2. Food intake was decreased during treatment in 3/8 cats in the placebo group and 4/8 cats treated with the synbiotic. Food intake was significantly associated with week of treatment ($F$-value 5.2, $P < 0.01$) but did not differ between treatment groups ($P = 0.5$). Percent days vomiting per week was significantly associated with week of treatment ($F$-value 8.6, $P < 0.01$) but not treatment group ($P = 0.4$). It was highest in the first week of treatment, during which 14/16 cats had at least 1 episode of vomiting, seven in each group. Fecal scores increased significantly during treatment ($F$-value 9.4, $P < 0.01$) but did not differ between treatment groups ($P = 0.2$). All cats in each treatment group had fecal scores ≥5 at some point during treatment, although high daily variability was noted. There was no significant effect of age, weight, or sex on any of the clinical parameters evaluated.

## Fecal microbiome

Cats in both treatment groups had marked changes in their fecal microbiome over time (Figs. 2–3). Alpha diversity was significantly lower ($P < 0.01$) on days 26–28, based on the Shannon index, goods coverage, and Chao1 metric (Table 3). There was no significant association between treatment group and changes in Shannon index, goods coverage, or the Chao1 metric. Similarly, beta diversity was significantly altered on days 26–28 compared to baseline and days 631–633 in each treatment group ($P < 0.01$; $R = 0.73$–0.76) with no significant differences between treatment groups (Fig. 2). The dysbiosis index was

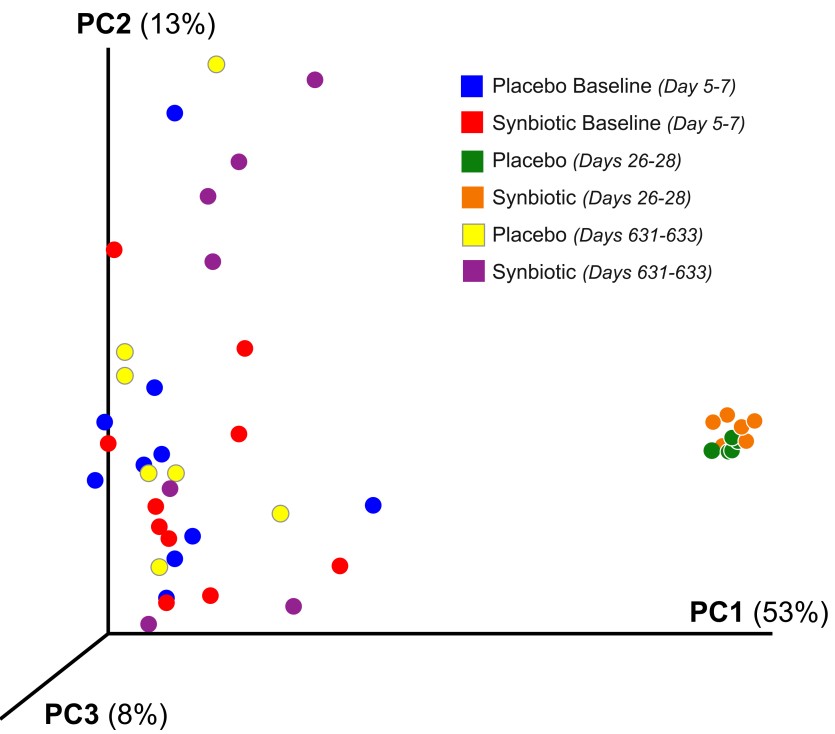

**Figure 2  Principal Coordinate Analysis (PCoA) of unweighted UniFrac distances of 16S rRNA genes.**
Gene sequences were determined using fecal samples collected at baseline (days 5–7), at the conclusion of
antibiotic administration (days 26–28), and after a 603 day washout (days 631–633) from 16 healthy cats,
eight per group, [+] that received 150 mg clindamycin with either a placebo or synbiotic PO once daily for
21 days. [+]Feces from one cat (synbiotic group) unavailable on days 631–633.

significantly higher on days 26–28 than at baseline or on days 631–633 in each treatment
group (Table 3), and a significant association was found between treatment group and
dysbiosis index (Table 3, *F*-value 13.6, *P* < 0.01).

Five phyla were identified based on sequencing analysis (mean baseline prevalences):
*Actinobacteria* (67.95%), *Firmicutes* (30.39%), *Proteobacteria* (1.03%), *Bacteroidetes*
(0.73%), and *Fusobacteria* (0.02%), with marked and significant differences over time
in relative abundances of all phyla except *Fusobacteria* (Fig. 3, Table 4). There was no
significant effect of treatment group on changes in relative bacterial abundances at any
timepoint, and there was no significant association between age or sex and changes in any
of the analyzed fecal microbiome parameters.

## Metabolome

Untargeted fecal metabolomics profiles differed markedly over time for cats in both
treatment groups (Figs. 4–5), with a variety of patterns of changes found. Fecal profiles
for 178 of 252 assayed compounds (71%) differed significantly over time (see Additional
file 1). Of those that differed over time, 47 of 178 (26.4%) metabolites had equivalent
profiles at baseline and days 631–633. Partial return to baseline profiles were identified
for an additional 14 metabolites (7.9%), and 21 metabolites (11.8%) had altered profiles

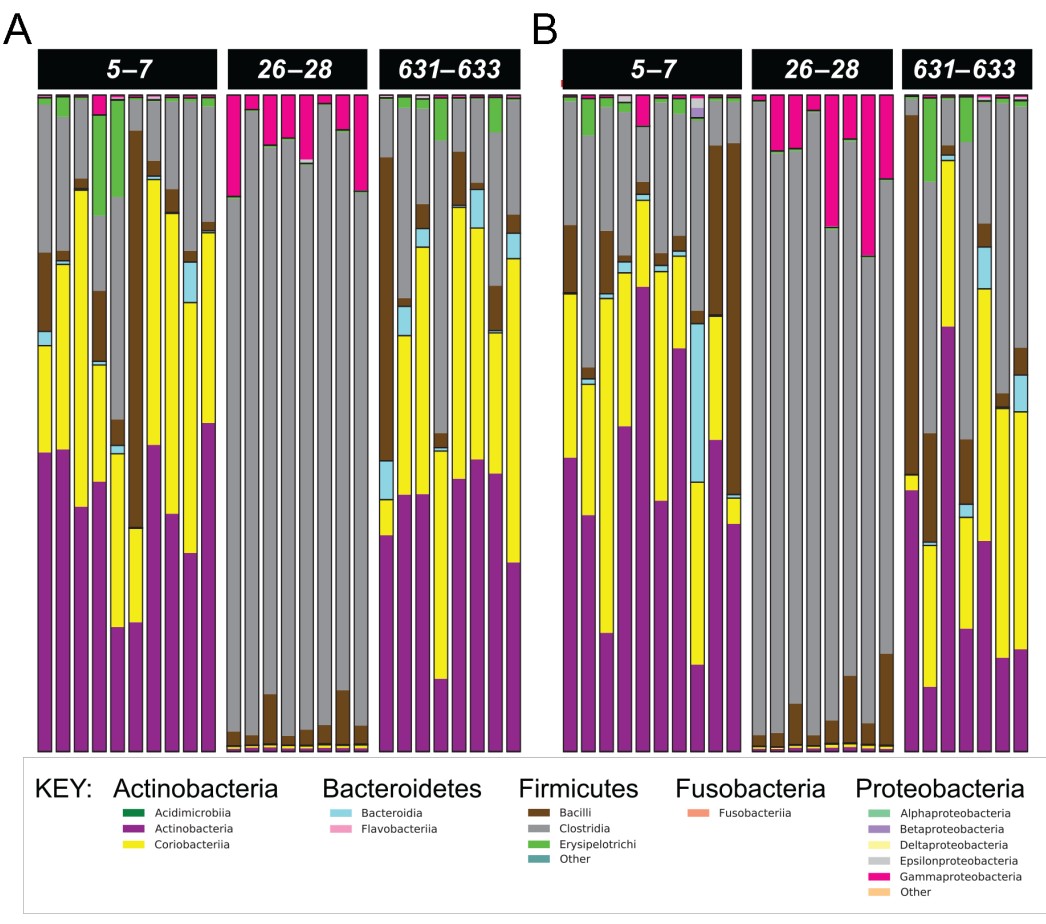

**Figure 3  Phylum- and class-level composition of fecal microbiota obtained from feline fecal samples.** (A) Placebo. (B) Synbiotic. Samples were collected at baseline (days 5–7), at the conclusion of antibiotic administration (days 26–28), and after a 603 day washout (days 631–633) from 16 healthy cats, eight per group, [+] that received 150 mg clindamycin with either a placebo or synbiotic PO once daily for 21 days. [+]Feces from one cat (synbiotic group) unavailable on days 631–633. Legend for all detected classes is shown, grouped by phylum.

on days 26–28 compared to baseline with no significant changes between days 26–28 and days 631–633. Changes in profiles for the remaining 96 metabolites (53.9%) were more complex. Profiles for 33 compounds (18.5%) differed significantly on days 631–633 from both baseline and days 26–28 with no difference between baseline and days 26–28, and profiles for 14 metabolites (7.9%) were significantly higher or lower at days 26–28 compared to baseline with further significant changes in the same direction by days 631–633. In contrast, profiles for 30 metabolites (16.9%) were significantly higher at days 26–28 than baseline but also significantly lower at days 631–633 than baseline; profiles for another 14 metabolites (7.9%) were decreased on days 26–28 with much higher profiles on days 631–633 than baseline. Finally, 5 metabolites had individual patterns of change.

Significant differences (fdr adjusted $P < 0.05$) were noted between treatment groups for seven metabolite profiles (Fig. 6). Significant group by time interactions were found for

**Table 3 Alpha diversity and dysbiosis index results for cats that received clindamycin and placebo or synbiotic.** Median (range) results for feces collected at baseline (days 5–7), at the conclusion of antibiotic administration (days 26–28), and after a 603 day washout (days 631–633) from 16 healthy cats, eight per group, + that received 150 mg clindamycin with either a placebo or synbiotic PO once daily for 21 days. + Feces from one cat (synbiotic group) unavailable at the days 631–633 timepoint. Cells that do not share a common superscript letter differed significantly ($P < 0.05$) based on post-hoc analysis.

| | Baseline | | Days 26–28 | | Days 631–633 | | P-value |
|---|---|---|---|---|---|---|---|
| | Placebo | Synbiotic | Placebo | Synbiotic | Placebo | Synbiotic | |
| Shannon index | 3.9[a] (3.3–5.3) | 4.2[a] (3.5–4.8) | 2.5[b] (1.8–3.0) | 2.8[b] (2.1–3.1) | 4.4[a] (3.6–5.2) | 4.9[a] (3.0–5.6) | <0.001 |
| Goods coverage | 0.9859[a] (0.9794–0.9919) | 0.9857[a] (0.9819–0.9920) | 0.9906[b] (0.9870–0.9942) | 0.9907[b] (0.9875–0.9927) | 0.9858[a] (0.9809–0.9909) | 0.9866[a] (0.9815–0.9894) | <0.001 |
| Chao1 metric | 2,085[a] (1,126–3,158) | 1,960[a] (1,044–2,930) | 1,419[b] (830–2,026) | 1,408[b] (1,043–2,062) | 2,119[a] (1,272–2,945) | 1,976[a] (1,349–2,622) | <0.001 |
| Dysbiosis index | −2.9[c] (−5.5 to −1.6) | −3.0[c] (−5.2 to −0.2) | 1.6[b] (0.4 to 3.3) | 4.7[a] (1.1 to 5.4) | −2.6[c] (−5.1 to −1.0) | −4.4[c] (−5.2 to −0.4) | <0.001 |

fecal profiles of putrescine, isopentadecanoic acid, cellobiose, ethanolamine, lactose, and D-erythro-sphingosine, while fecal profiles for N-acetylglutamate differed significantly by treatment group alone.

With regard to metabolites of known biological importance, there were significant changes in profiles of some short-chain fatty acid (SCFA) synthesis metabolites, bile acids, tryptophan degradation pathway metabolites, sphingolipids, polyamines, and benzoic and cinnaminic acids over time (Table 5). Pathway analysis revealed significant changes (fdr adjusted $P < 0.05$) over time for 55 metabolic pathways; 47 of which had impact factors >0 (Table 6). The majority of pathways affected were related to SCFA, amino acid, sugar, and nucleotide synthesis and degradation. Amino acid pathways that were affected included those related to tryptophan degradation. Other pathways significantly affected included those related to linoleic acid, sphingolipid, and glycerolipid metabolism.

## DISCUSSION

The incidence of AAGS varies in people based on individual antibiotic agent, prior antibiotic exposure, subject age, and other factors (*Hempel et al., 2012*; *Lenoir-Wijnkoop et al., 2014*; *McFarland, 2008*; *Ouwehand et al., 2016*). Antibiotics, such as clindamycin, can cause AAGS directly by stimulating the intestinal motilin receptor (*Bartlett, 2002*). However, antibiotic-induced dysbiosis, with secondary alterations in fecal metabolites and overgrowth of opportunistic pathogens, is believed to be the primary contributor to AAGS (*Gustafsson et al., 1998*; *McFarland, 2008*; *Varughese, Vakil & Phillips, 2013*; *Videlock & Cremonini, 2012*). Dysbiosis is greatest for broad-spectrum and poorly-absorbed antibiotics, including macrolides, cephalosporins, and fluoroquinolones (*Jakobsson et al., 2010*; *Löfmark et al., 2006*; *McFarland, 2008*; *Ouwehand et al., 2016*), and alterations to the microbiome can persist at least 4 years after short-term antibiotic therapy (*Jakobsson et al., 2010*; *Löfmark et al., 2006*).

Multiple metanalyses support the efficacy of probiotics in preventing AAGS in people (*Goldenberg et al., 2015*; *Lau & Chamberlain, 2016*; *McFarland, 2016*; *Ouwehand et al.,*

**Table 4  Median percent abundance (range) for taxa with significantly different relative abundances over time.** Median (range) percent relative abundances of different taxa in feces were collected at baseline (days 5–7), at the conclusion of antibiotic administration (days 26–28), and after a 603 day washout (days 631–633) from 16 healthy cats, eight per group, + that received 150 mg clindamycin with either a placebo or synbiotic PO once daily for 21 days. + Feces from one cat (synbiotic group) unavailable at the days 631–633 timepoint.

| | Baseline | | Days 26–28 | | Days 631–633 | | Fdr-adjusted *P*-value[*] |
|---|---|---|---|---|---|---|---|
| | Placebo | Synbiotic | Placebo | Synbiotic | Placebo | Synbiotic | |
| *Actinobacteria* | 68.02[a] (34.02–87.12) | 71.32[a] (55.95–83.97) | 0.96[b] (0.82–1.07) | 0.92[b] (0.69–1.20) | 69.43[a] (38.38–82.91) | 51.75[a] (31.39–90.03) | <0.001 |
| ●*Actinobacteria* | 68.02[a] (34.02–87.12) | 71.32[a] (55.95–83.97) | 0.96[b] (0.82–1.07) | 0.92[b] (0.69–1.20) | 69.43[a] (38.38–82.91) | 51.75[a] (31.39–90.03) | <0.001 |
| −*Actinomycetales* | 0.04[a] (0.00–0.07) | 0.04[a] (0.02–0.06) | 0.00[b] (0.00–0.01) | 0.00[b] (0.00–0.01) | 0.03[a] (0.01–0.13) | 0.03[a] (0.02–0.16) | <0.001 |
| o *Actinomycetaceae* | 0.03[a] (0.00–0.06) | 0.04[a] (0.01–0.06) | 0.00[b] (0.00–0.01) | 0.00[b] (0.00–0.01) | 0.03[a] (0.01–0.13) | 0.03[a] (0.01–0.14) | <0.001 |
| ■ *Actinomyces* | 0.033[a] (0.000–0.060) | 0.037[a] (0.009–0.060) | 0.001[b] (0.000–0.006) | 0.003[b] (0.000–0.006) | 0.026[a] (0.006–0.126) | 0.034[a] (0.014–0.143) | <0.001 |
| −*Bifidobacteriales* | 39.16[a] (18.91–46.70) | 46.10[a] (18.05–70.73) | 0.54[b] (0.45–0.65) | 0.49[b] (0.42–0.71) | 39.16[c] (10.98–44.47) | 18.72[c] (9.85–64.69) | <0.001 |
| o *Bifidobacteriaceae* | 39.16[a] (18.91–46.70) | 46.10[a] (18.05–70.73) | 0.54[b] (0.45–0.65) | 0.49[b] (0.42–0.71) | 39.16[c] (10.98–44.47) | 18.72[c] (9.85–64.69) | <0.001 |
| ■ *Bifidobacterium* | 39.12[a] (18.91–46.70) | 46.08[a] (18.01–70.72) | 0.54[b] (0.45–0.65) | 0.49[b] (0.42–0.71) | 39.12[c] (10.97–44.46) | 18.70[c] (9.82–64.66) | <0.001 |
| ●*Coriobacteriia* | 27.29[a] (14.31–48.21) | 21.64[a] (13.19–50.92) | 0.42[b] (0.32–0.48) | 0.40[b] (0.26–0.53) | 34.95[a] (5.43–46.25) | 25.28[a] (2.33–38.40) | <0.001 |
| −*Coriobacteriales* | 27.29[a] (14.31–48.21) | 21.64[a] (13.19–50.92) | 0.42[b] (0.32–0.48) | 0.40[b] (0.26–0.53) | 34.95[a] (5.43–46.25) | 25.28[a] (2.33–38.40) | <0.001 |
| o *Coriobacteriaceae* | 27.29[a] (14.31–48.21) | 21.64[a] (13.19–50.92) | 0.42[b] (0.32–0.48) | 0.40[b] (0.26–0.53) | 34.95[a] (5.43–46.25) | 25.28[a] (2.33–38.40) | <0.001 |
| ■ *Adlercreutzia* | 0.024[a] (0.003–0.051) | 0.014[a] (0.006–0.037) | 0.003[b] (0.000–0.003) | 0.000[b] (0.000–0.003) | 0.059[c] (0.006–0.094) | 0.031[c] (0.009–0.217) | <0.001 |
| ■ *Collinsella* | 9.39[a] (1.39–25.36) | 8.91[a] (3.19–21.39) | 0.21[b] (0.17–0.24) | 0.21[b] (0.16–0.25) | 8.69[a] (4.17–33.50) | 16.18[a] (1.88–20.90) | <0.001 |
| ■ *Slackia* | 0.13[a] (0.04–0.21) | 0.11[a] (0.05–0.27) | 0.01[b] (0.00–0.02) | 0.01[b] (0.00–0.01) | 0.17[a] (0.14–0.30) | 0.19[a] (0.01–0.35) | <0.001 |
| *Bacteroidetes* | 0.54[a] (0.11–2.17) | 0.80[a] (0.17–1.67) | 0.09[b] (0.05–0.15) | 0.10[b] (0.05–0.13) | 3.35[a] (0.30–5.91) | 0.82[a] (0.08–6.37) | <0.001 |
| ●*Bacteroidia* | 0.54[a] (0.11–2.17) | 0.80[a] (0.17–1.67) | 0.09[b] (0.05–0.15) | 0.10[b] (0.05–0.13) | 3.35[a] (0.30–5.91) | 0.81[a] (0.08–6.37) | <0.001 |
| −*Bacteroidales* | 0.54[a] (0.11–2.17) | 0.80[a] (0.17–1.67) | 0.09[b] (0.05–0.15) | 0.10[b] (0.05–0.13) | 3.35[a] (0.30–5.91) | 0.81[a] (0.08–6.37) | <0.001 |
| o *Bacteroidaceae* | 0.14[a] (0.07–1.67) | 0.24[a] (0.10–1.13) | 0.07[b] (0.04–0.13) | 0.09[b] (0.04–0.11) | 3.08[c] (0.21–5.85) | 0.53[c] (0.07–6.03) | <0.001 |
| ■ *Bacteroides* | 0.14[a] (0.07–1.67) | 0.24[a] (0.10–1.13) | 0.07[b] (0.04–0.13) | 0.09[b] (0.04–0.11) | 3.08[c] (0.21–5.85) | 0.53[c] (0.07–6.03) | <0.001 |

*(continued on next page)*
**Table 4** (*continued*)

| | Baseline | | Days 26–28 | | Days 631–633 | | Fdr-adjusted P-value[*] |
|---|---|---|---|---|---|---|---|
| | Placebo | Synbiotic | Placebo | Synbiotic | Placebo | Synbiotic | |
| o *Porphyromonadaceae* | 0.007[a] (0.003–0.349) | 0.013[a] (0.006–0.037) | 0.003[b] (0.003–0.011) | 0.006[b] (0.003–0.017) | 0.071[c] (0.014–0.514) | 0.063[c] (0.003–0.631) | <0.001 |
| ■ *Parabacteroides* | 0.007[a] (0.003–0.349) | 0.013[a] (0.006–0.037) | 0.003[b] (0.003–0.011) | 0.006[b] (0.003–0.017) | 0.071[c] (0.014–0.514) | 0.063[c] (0.003–0.631) | <0.001 |
| o *Prevotellaceae* | 0.19[a] (0.01–0.63) | 0.36[a] (0.02–0.79) | 0.009[b] (0.003–0.014) | 0.006[b] (0.000–0.011) | 0.10[a] (0.01–0.21) | 0.14[a] (0.01–0.28) | <0.001 |
| ■ *Prevotella* | 0.194[a] (0.011–0.634) | 0.363[a] (0.020–0.794) | 0.009[b] (0.003–0.014) | 0.006[b] (0.000–0.011) | 0.100[a] (0.014–0.206) | 0.137[a] (0.006–0.277) | <0.001 |
| ***Firmicutes*** | 30.29[a] (11.58–65.48) | 27.64[a] (10.37–42.67) | 91.84[b] (83.66–97.50) | 90.75[b] (74.36–98.31) | 26.21[a] (13.98–55.26) | 47.08[a] (8.84–67.65) | <0.001 |
| ●*Clostridia* | 12.69[a] (4.63–33.88) | 18.83[a] (6.76–35.33) | 85.49[b] (81.13–95.00) | 82.78[b] (70.84–96.36) | 15.95[a] (7.95–44.59) | 36.75[a] (2.44–45.25) | <0.001 |
| −*Clostridiales* | 12.69[a] (4.63–33.88) | 18.83[a] (6.76–35.33) | 85.49[b] (81.13–95.00) | 82.78[b] (70.84–96.36) | 15.95[a] (7.95–44.59) | 36.75[a] (2.44–45.25) | <0.001 |
| o *Clostridiales*; Other | 0.25[a] (0.10–1.14) | 0.31[a] (0.05–0.97) | 0.06[b] (0.03–0.13) | 0.05[b] (0.03–0.13) | 0.79[a] (0.09–2.01) | 0.54[a] (0.01–2.47) | <0.001 |
| ■ Other | 0.25[a] (0.10–1.14) | 0.31[a] (0.05–0.97) | 0.06[b] (0.03–0.13) | 0.05[b] (0.03–0.13) | 0.79[a] (0.09–2.01) | 0.54[a] (0.01–2.47) | <0.001 |
| o *Clostridiaceae* | 1.97[a] (1.32–6.98) | 2.28[a] (1.56–11.82) | 82.66[b] (78.23–92.91) | 80.16[b] (68.32–94.47) | 3.56[a] (1.23–22.91) | 5.99[a] (1.12–24.29) | <0.001 |
| ■ Other | 0.77[a] (0.47–2.22) | 0.75[a] (0.47–4.83) | 66.32[b] (60.30–77.61) | 57.64[b] (54.96–72.51) | 0.67[a] (0.48–2.33) | 0.82[a] (0.44–5.27) | <0.001 |
| ■ ___ | 0.83[a] (0.57–5.43) | 1.40[a] (0.82–10.25) | 8.52[b] (3.79–12.68) | 7.67[b] (3.53–13.62) | 2.71[c] (0.65–18.51) | 4.76[c] (0.47–21.95) | <0.001 |
| ■ *Clostridium* | 0.16[a] (0.11–0.25) | 0.16[a] (0.08–0.47) | 9.97[b] (8.78–11.13) | 9.20[b] (7.90–11.19) | 0.15[a] (0.09–1.63) | 0.19[a] (0.11–0.55) | <0.001 |
| ■ SMB53 | 0.020[a] (0.009–0.151) | 0.026[a] (0.014–0.389) | 0.011[b] (0.009–0.026) | 0.016[b] (0.011–0.077) | 0.089[c] (0.014–0.740) | 0.197[c] (0.009–0.817) | <0.001 |
| o *Lachnospiraceae* | 6.71[a] (1.66–16.53) | 9.12[a] (1.05–14.87) | 1.40[b] (1.14–1.79) | 1.46[b] (1.06–1.74) | 7.92[a] (2.47–16.14) | 14.56[a] (0.75–20.25) | <0.001 |
| ■ Other | 0.33[a] (0.05–0.73) | 0.68[a] (0.07–1.50) | 0.06[b] (0.04–0.11) | 0.06[b] (0.03–0.09) | 0.77[a] (0.27–1.28) | 0.78[a] (0.05–1.97) | <0.001 |
| ■ *Blautia* | 4.75[a] (1.16–10.51) | 6.16[a] (0.42–8.29) | 0.58[b] (0.48–0.69) | 0.60[b] (0.46–0.73) | 4.49[a] (1.39–12.61) | 10.91[a] (0.36–14.02) | 0.002 |
| ■ *Coprococcus* | 0.016[a] (0.011–0.066) | 0.027[a] (0.011–0.091) | 0.014[b] (0.009–0.031) | 0.013[b] (0.003–0.023) | 0.050[c] (0.014–0.117) | 0.086[c] (0.000–0.203) | 0.002 |
| ■ *Dorea* | 0.52[a] (0.20–1.40) | 0.54[a] (0.16–0.96) | 0.35[a] (0.30–0.47) | 0.33[a] (0.24–0.44) | 0.69[b] (0.31–1.41) | 1.01[b] (0.18–2.23) | 0.014 |
| ■ *Roseburia* | 0.011[a] (0.003–0.151) | 0.013[a] (0.003–0.040) | 0.007[a] (0.000–0.020) | 0.009[a] (0.000–0.023) | 0.031[b] (0.026–0.160) | 0.040[b] (0.003–0.280) | <0.001 |
| ■ [*Ruminococcus*] | 0.92[a] (0.17–5.49) | 1.14[a] (0.24–3.84) | 0.25[b] (0.20–0.38) | 0.28[b] (0.18–0.35) | 0.74[a] (0.21–1.20) | 1.19[a] (0.11–2.05) | 0.002 |
| o *Peptococcaceae* | 0.15[a] (0.06–0.40) | 0.20[a] (0.03–0.71) | 0.06[b] (0.02–0.09) | 0.06[b] (0.03–0.09) | 0.23[a] (0.04–1.68) | 0.34[a] (0.01–1.67) | 0.015 |
| ■ *Peptococcus* | 0.15[a] (0.06–0.40) | 0.20[a] (0.03–0.71) | 0.06[b] (0.02–0.09) | 0.06[b] (0.03–0.09) | 0.23[a] (0.04–1.68) | 0.34[a] (0.01–1.67) | 0.014 |

**Table 4 (*continued*)**

| | Baseline | | Days 26–28 | | Days 631–633 | | Fdr-adjusted *P*-value[*] |
|---|---|---|---|---|---|---|---|
| | Placebo | Synbiotic | Placebo | Synbiotic | Placebo | Synbiotic | |
| o *Ruminococcaceae* | 1.22[a] (0.25–3.55) | 1.97[a] (0.14–9.85) | 0.25[b] (0.18–0.51) | 0.22[b] (0.17–0.38) | 0.82[a] (0.16–4.79) | 0.58[a] (0.11–2.54) | <0.001 |
| ■ *Faecalibacterium* | 0.010[a] (0.000–0.146) | 0.031[a] (0.003–0.077) | 0.003[b] (0.000–0.009) | 0.001[b] (0.000–0.011) | 0.006[a] (0.000–0.063) | 0.011[a] (0.000–0.091) | 0.016 |
| ■ *Oscillospira* | 0.029[a] (0.009–0.163) | 0.033[a] (0.011–0.057) | 0.127[b] (0.066–0.320) | 0.100[b] (0.023–0.174) | 0.083[b] (0.026–0.434) | 0.160[b] (0.006–0.543) | <0.001 |
| ■ *Ruminococcus* | 0.017[a] (0.006–0.031) | 0.017[a] (0.006–0.040) | 0.009[b] (0.006–0.017) | 0.010[b] (0.003–0.014) | 0.034[c] (0.011–0.160) | 0.037[c] (0.003–0.054) | 0.002 |
| o *Veillonellaceae* | 1.03[a] (0.13–4.81) | 0.97[a] (0.54–4.88) | 0.07[b] (0.03–0.12) | 0.08[b] (0.03–0.16) | 0.26[c] (0.08–2.63) | 0.17[c] (0.08–3.91) | <0.001 |
| ■ *Megamonas* | 0.11[a] (0.01–0.22) | 0.21[a] (0.02–2.38) | 0.02[b] (0.00–0.05) | 0.02[b] (0.01–0.06) | 0.04[a] (0.02–0.35) | 0.05[a] (0.00–0.28) | 0.009 |
| ■ *Megasphaera* | 0.86[a] (0.08–4.68) | 0.72[a] (0.08–2.42) | 0.04[b] (0.02–0.07) | 0.05[b] (0.02–0.08) | 0.23[c] (0.05–2.20) | 0.09[c] (0.04–3.58) | <0.001 |
| ■ *Phascolarctobacterium* | 0.009[a] (0.003–0.029) | 0.009[a] (0.000–0.069) | 0.000[b] (0.000–0.006) | 0.006[b] (0.000–0.017) | 0.011[a] (0.003–0.063) | 0.003[a] (0.000–0.046) | 0.004 |
| ● *Erysipelotrichi* | 0.81[a] (0.21–15.32) | 1.05[a] (0.18–5.59) | 0.28[b] (0.21–0.36) | 0.30[b] (0.22–0.37) | 0.89[a] (0.21–6.46) | 0.78[a] (0.24–12.78) | 0.008 |
| − *Erysipelotrichiales* | 0.81[a] (0.21–15.32) | 1.05[a] (0.18–5.59) | 0.28[b] (0.21–0.36) | 0.30[b] (0.22–0.37) | 0.89[a] (0.21–6.46) | 0.78[a] (0.24–12.78) | 0.005 |
| o *Erysipelotrichaceae* | 0.81[a] (0.21–15.32) | 1.05[a] (0.18–5.59) | 0.28[b] (0.21–0.36) | 0.30[b] (0.22–0.37) | 0.89[a] (0.21–6.46) | 0.78[a] (0.24–12.78) | 0.006 |
| ■ *Bulleidia* | 0.019[a] (0.006–0.194) | 0.006[a] (0.000–0.123) | 0.003[b] (0.000–0.009) | 0.004[b] (0.000–0.011) | 0.016[a] (0.000–0.431) | 0.009[a] (0.003–0.014) | 0.015 |
| ■ *[Eubacterium]* | 0.26[a] (0.08–6.60) | 0.22[a] (0.04–1.17) | 0.07[b] (0.05–0.12) | 0.08[b] (0.03–0.13) | 0.27[a] (0.08–2.94) | 0.33[a] (0.10–6.63) | <0.001 |
| *Proteobacteria* | 0.39[a] (0.24–3.03) | 0.48[a] (0.29–4.74) | 7.09[b] (1.28–15.46) | 8.34[b] (0.85–24.53) | 0.43[a] (0.28–0.57) | 0.43[a] (0.21–0.89) | <0.001 |
| ● *Gammaproteobacteria* | 0.27[a] (0.16–2.98) | 0.34[a] (0.18–4.63) | 7.04[b] (1.23–15.32) | 8.30[b] (0.83–24.39) | 0.25[c] (0.14–0.37) | 0.31[c] (0.18–0.37) | <0.001 |
| − *Aeromonadales* | 0.01[a] (0.00–0.07) | 0.02[a] (0.01–0.04) | 0.03[b] (0.02–0.07) | 0.03[b] (0.01–0.05) | 0.01[a] (0.00–0.03) | 0.01[a] (0.01–0.04) | 0.010 |
| o *Succinivibrionaceae* | 0.009[a] (0.003–0.069) | 0.021[a] (0.006–0.043) | 0.033[b] (0.017–0.071) | 0.030[b] (0.014–0.051) | 0.014[a] (0.003–0.029) | 0.014[a] (0.006–0.040) | 0.013 |
| ■ *Anaerobiospirillumi* | 0.007[a] (0.003–0.060) | 0.017[a] (0.006–0.029) | 0.031[b] (0.017–0.071) | 0.030[b] (0.014–0.051) | 0.014[a] (0.003–0.023) | 0.011[a] (0.006–0.037) | 0.002 |
| − *Enterobacteriales* | 0.26[a] (0.15–2.92) | 0.32[a] (0.14–4.62) | 6.99[b] (1.19–15.30) | 8.25[b] (0.81–24.34) | 0.23[c] (0.13–0.34) | 0.28[c] (0.17–0.33) | <0.001 |
| o *Enterobacteriaceae* | 0.26[a] (0.15–2.92) | 0.32[a] (0.14–4.62) | 6.99[b] (1.19–15.30) | 8.25[b] (0.81–24.34) | 0.23[c] (0.13–0.34) | 0.28[c] (0.17–0.33) | <0.001 |

**Notes.**

[*]*P*-values adjusted based on the Benjamini and Hochberg False discovery rate (fdr).

Relative abundances that do not share a common superscript letter differed significantly (fdr-adjusted *P* < 0.05) based on post-hoc analysis.

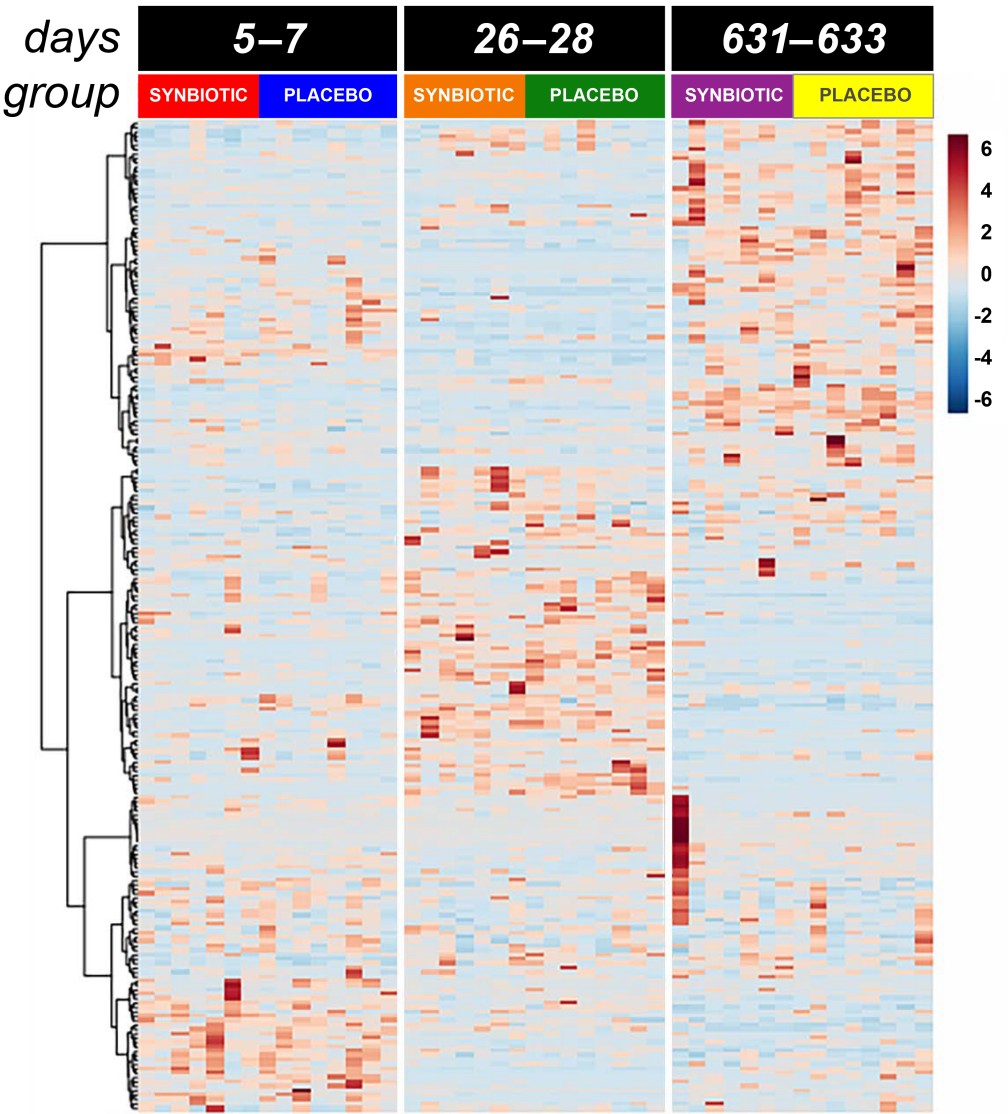

**Figure 4** **Dual hierarchical dendrogram of metabolites, clustered by pathway, that differed significantly over time in feline fecal samples.** Samples were collected at baseline (days 5–7), at the conclusion of antibiotic administration (days 26–28), and after a 603 day washout (days 631–633) from 16 healthy cats, 8 per group, [+] that received 150 mg clindamycin with either a placebo or synbiotic PO once daily for 21 days. [+] Feces from one cat (synbiotic group) unavailable on days 631–633.

2016; *Szajewska, Konarska & Kołodziej, 2016*; *Videlock & Cremonini, 2012*). Although some gastrointestinal effects appear to be dose-related (*Ouwehand et al., 2016*; *Pagnini et al., 2010*; *Szajewska, Konarska & Kołodziej, 2016*), effects do not correlate directly with the relative abundances of probiotic bacteria or measurable shifts in OTUs in the fecal microbiome (*Biagi et al., 2013*; *Garcia-Mazcorro et al., 2011*; *Marshall-Jones et al., 2006*; *Rossi et al., 2014*). The ideal microbial strain(s), dose, and timing relative to antibiotic administration likely vary by antibiotic, underlying disease processes,

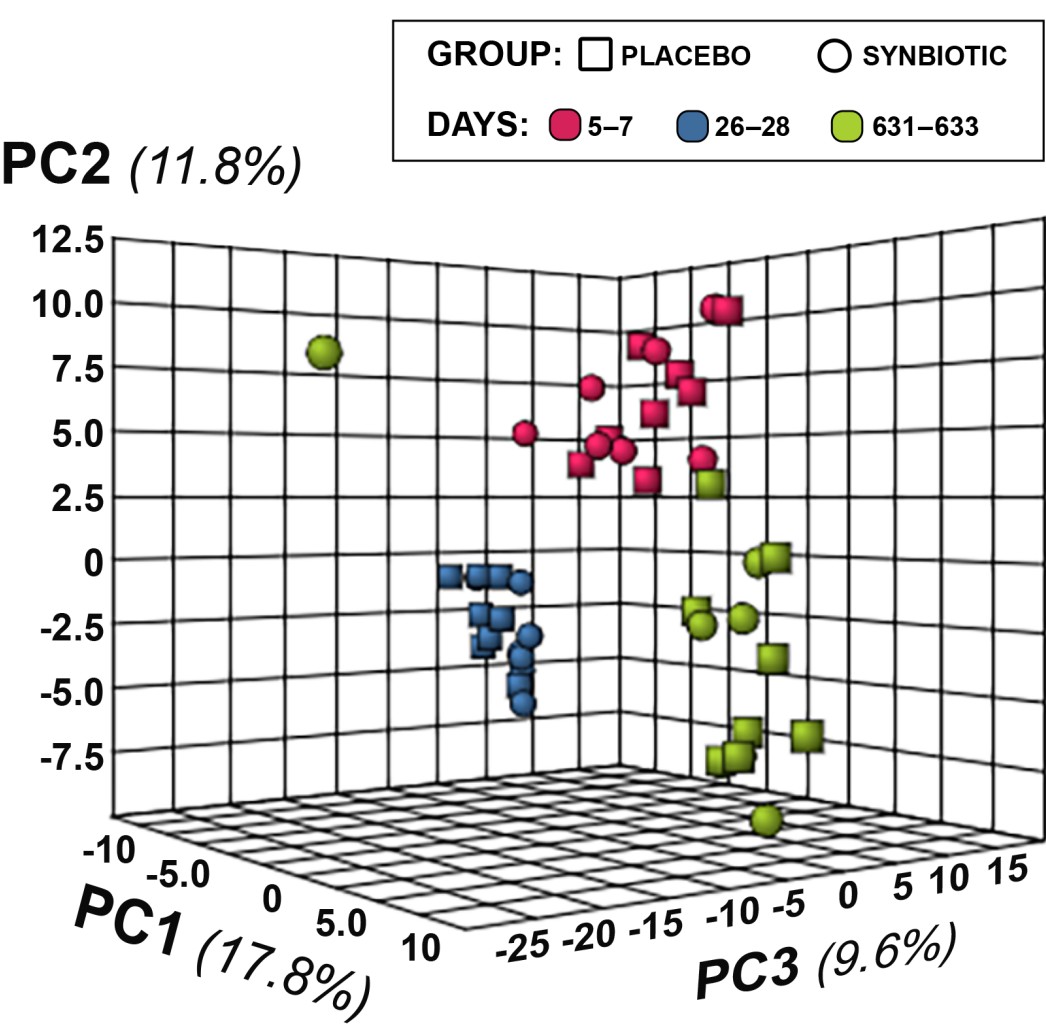

**Figure 5** **Principal Component Analysis (PCA) of metabolic pathway analyses from feline fecal samples.** Samples were collected at baseline (days 5–7), at the conclusion of antibiotic administration (days 26–28), and after a 603 day washout (days 631–633) from 16 healthy cats, eight per group, [+] that received 150 mg clindamycin with either a placebo or synbiotic PO once daily for 21 days. [+]Feces from one cat (synbiotic group) unavailable on days 631–633.

patient (age, inpatient status, historical antibiotic exposure), and other factors (*Lau & Chamberlain, 2016*; *McFarland, 2009*; *Szajewska, Konarska & Kołodziej, 2016*). Positive effects of probiotics include stimulation of the host's innate system and epithelial secretion of antimicrobial substances, restoration of intestinal barrier function, alteration of the gastrointestinal metabolome, disruption of biofilms through production of bacteriocins, generation of other antimicrobial agents, and competitive exclusion of pathogens (*Ohland & MacNaughton, 2010*; *Ouwehand et al., 2016*; *Pagnini et al., 2010*; *Quigley, 2012*; *Rossi et al., 2014*; *Strompfová, Lauková & Ouwehand, 2004*).

Although AAGS are a recognized complication of antibiotic therapy in cats (*Albarellos & Landoni, 2009*; *Bureau of Veterinary Drugs, 1995*; *Hunter et al., 1995*), the incidence of

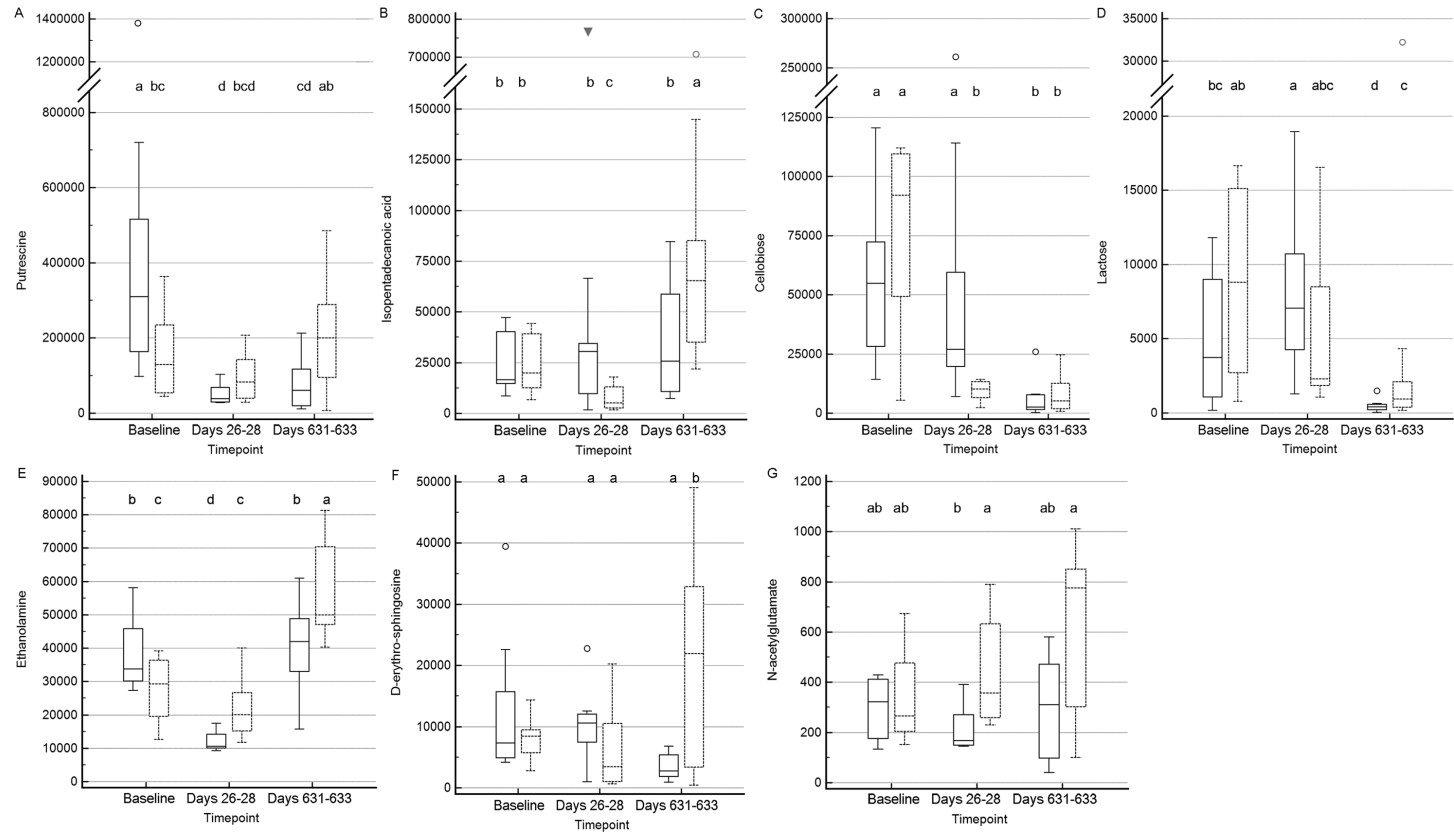

**Figure 6** Box and whisker plots of fecal metabolite profiles for seven fecal metabolites that differed significantly (fdr adjusted $P < 0.05$) between treatment groups and over time. (A) Putrescine. (B) Isopentadecanoic acid. (C) Cellobiose. (D) Lactose. (E) Ethanolamine. (F) D-erythro-sphingosine. (G) N-acetylglutamate. Medians, interquartile ranges, and minimum and maximum values are presented for cats in the placebo (boxes with solid borders) and synbiotic (boxes with dashed borders) treatment groups. Open circles and closed triangles denote outlier values. Boxes that do not share a letter differed significantly (fdr-adjusted $P < 0.05$) based on post-hoc analysis. Fecal samples were collected at baseline (days 5–7), at the conclusion of antibiotic administration (days 26–28), and after a 603 day washout (days 631–633 from 16 healthy cats, 8 per group, [+] that received 150 mg clindamycin with either a placebo or synbiotic PO once daily for 21 days. [+]Feces from one cat (synbiotic group) unavailable on days 631–633. Significance was set as $P < 0.05$, with $P$- values adjusted based on the Benjamini and Hochberg False discovery rate (fdr).

AAGS, short- and long-term effects on the microbiome and metabolome, and impact of concurrent synbiotic administration had not been previously characterized. In this study, AAGS developed in 100% of cats treated with high dose clindamycin once daily for 21 days. Vomiting was most severe in the first week of therapy, although it persisted for the duration of treatment in some cats. A temporal association between antibiotic administration and vomiting was not noted, suggesting stimulation of the motilin receptor by clindamycin had minimal impact on the incidence of vomiting. The incidence of vomiting per week was less in cats receiving the synbiotic, but the difference between treatment groups was not statistically significant. Fecal consistency and food intake declined progressively throughout the course of treatment in both treatment groups.

There are a number of potential explanations for the lack of statistically significant differences in AAGS between treatment groups in this study. A high antibiotic dose was used

Whittemore et al. (2018), *PeerJ*, DOI 10.7717/peerj.5130

**Table 5  Metabolites of known biological import with profiles that significantly differed over time.** Median (range) peak height of metabolites in feces collected at baseline (days 5–7), at the conclusion of antibiotic administration (days 26–28), and after a 603 day washout (days 631–633) from 16 healthy cats, eight per group, + that received 150 mg clindamycin with either a placebo or synbiotic PO once daily for 21 days. + Feces from one cat (synbiotic group) unavailable at days 631–633.

| | Baseline | | Days 26–28 | | Days 631–633 | | Fdr-adjusted *P*-value[*] |
|---|---|---|---|---|---|---|---|
| | Placebo | Synbiotic | Placebo | Synbiotic | Placebo | Synbiotic | |
| **Short chain fatty acid metabolites** | | | | | | | |
| 3,4-dihydroxyphenylacetic acid | 589[a] (317–1,201) | 521[a] (221–1,577) | 741[a] (263–1,130) | 594[a] (126–1,522) | 197[b] (66–1,436) | 199[b] (63–643) | <0.001 |
| 3-hydroxyphenylacetic acid | 3,350[a] (582–12,188) | 2,860[a] (55–11,401) | 65[b] (36–94) | 57[b] (26–239) | 554[c] (77–3,578) | 275[c] (124–758) | <0.001 |
| 4-hydroxyphenylacetic acid | 24,894[a] (10,957–34,200) | 24,248[a] (12,980–73,736) | 693[b] (473–6,935) | 3,268[b] (378–139,343) | 48,775[a] (10,270–133,645) | 47,322[a] (837–128,116) | <0.001 |
| 2,3-dihydroxybutanoic acid | 167[a] (119–347) | 193[a] (67–240) | 83[b] (18–157) | 108[b] (25–268) | 216[a] (97–356) | 128[a] (63–376) | <0.001 |
| 2-hydroxybutanoic acid | 12,514[a] (3,401–161,892) | 10,110[a] (626–61,119) | 38,839[b] (26,636–58,406) | 38,680[b] (18,670–76,553) | 3,846[c] (1,170–13,209) | 3,082[c] (623–13,862) | <0.001 |
| 4-aminobutyric acid | 2,002[a] (1,665–12,898) | 2,573[a] (1,189–11,658) | 112,677[b] (72,970–190,000) | 156,808[b] (56,870–393,600) | 26,744[c] (3,966–89,927) | 20,753[c] (437–60,568) | <0.001 |
| 2,4-diaminobutyric acid | 1,758[a] (568–3,102) | 1,119[a] (184–3,865) | 210[b] (140–350) | 254[b] (109–590) | 3,095[c] (1,652–8,818) | 3,017[c] (765–6,398) | <0.001 |
| 3-phenyllactic acid | 5,539[a] (781–31,484) | 5,796[a] (438–32,942) | 9,109[b] (7,089–11,018) | 12,813[b] (7,617–21,692) | 6,537[a] (1,217–19,572) | 9,052[a] (331–16,813) | <0.001 |
| Lactic acid | 152,579[a] (24,216–1,034,220) | 120,122[a] (10,433–1,403,821) | 392,894[b] (269,519–1,091,298) | 308,949[b] (129,795–638,477) | 27,338[c] (11,859–149,087) | 35,473[c] (6,439–111,809) | <0.001 |
| P-hydroxylphenyl-lactic acid | 3,698[a] (1,153–6,357) | 4,503[a] (581–12,311) | 3,590[a] (2,472–6,688) | 4,760[a] (2,300–5,993) | 1,654[b] (478–5,019) | 1,431[b] (405–2,022) | <0.001 |
| Propane-1-3-diol | 1,108[a] (426–15,239) | 1,900[a] (272–5,779) | 12,212[b] (3,684–26,363) | 12,461[b] (4,117–20,201) | 359[c] (174–1,203) | 341[c] (99–502) | <0.001 |
| 3,3-hydroxyphenyl propionic acid | 76,962[a] (18,783–212,470) | 45,560[a] (865–240,661) | 789[b] (88–1,348) | 648[b] (188–1,431) | 77,032[a] (666–167,768) | 27,581[a] (140–134,885) | <0.001 |
| 3-(4-hydroxyphenyl)-propionic acid | 65,664[a] (28,474–90,967) | 56,101[a] (4,122–141,022) | 4,255[b] (350–16,751) | 6,447[b] (415–16,156) | 28,844[c] (4,747–68,675) | 14,394[c] (12,923–79,425) | <0.001 |
| Pyruvic acid | 7,118[a] (1,844–19,684) | 7,900[a] (3,534–12,585) | 7,744[a] (3,418–20,148) | 9,650[a] (1,509–48,246) | 915[b] (456–3,736) | 1,588[b] (489–2,230) | <0.001 |
| Succinic acid | 236,185[a] (14,243–708,079) | 88,426[a] (882–1,122,477) | 41,400[a] (25,669–96,357) | 48,077[a] (7,555–107,389) | 8,572[b] (3,020–65,677) | 9,082[b] (1,337–65,270) | <0.001 |

Whittemore et al. (2018), *PeerJ*, DOI 10.7717/peerj.5130

**Table 5** (*continued*)

| | Baseline | | Days 26–28 | | Days 631–633 | | Fdr-adjusted *P*-value[*] |
|---|---|---|---|---|---|---|---|
| | **Placebo** | **Synbiotic** | **Placebo** | **Synbiotic** | **Placebo** | **Synbiotic** | |
| **Bile acids** | | | | | | | |
| Cholic acid | 109,466[a] (1,898–200,811) | 59,483[a] (423–220,771) | 11,303[b] (113–76,704) | 27,391[b] (286–89,000) | 9,511[b] (240–77,925) | 8,907[b] (3,191–95,296) | 0.04 |
| Deoxycholic acid | 3,295[a] (77–28,703) | 1,843[a] (109–43,646) | 136[b] (45–5,010) | 352[b] (27–3,531) | 11,544[a] (150–111,319) | 8,762[a] (687–309,022) | 0.01 |
| **Tryptophan metabolites** | | | | | | | |
| Indole-3-acetate | 13,713[a] (7,780–94,057) | 8,832[a] (3,110–125,618) | 841[b] (541–1,795) | 1,051[b] (448–11,089) | 15,203[a] (6,967–22,835) | 27,026[c] (17,204–168,850) | <0.001 |
| Indole-3-lactate | 80,846[a] (34,669–127,305) | 62,189[a] (7,536–128,029) | 6,751[b] (2,893–11,123) | 7,133[b] (4,437–22,490) | 15,933[b] (773–44,962) | 5,311[b] (836–54,443) | <0.001 |
| Tryptophan | 190,390[a] (132,057–340,750) | 173,956[a] (106,947–276,525) | 120,418[b] (95,182–158,327) | 153,738[b] (19,076–294,367) | 20,219[c] (3,724–138,699) | 35,369[c] (5,707–174,485) | <0.001 |
| **Sphingolipid metabolites** | | | | | | | |
| Cellobiose | 54,905[a] (14,235–120,530) | 92,176[a] (5,399–111,969) | 28,177[ab] (6,974–261,178) | 10,182[b] (2,284–14,269) | 2,513[b] (249–26,017) | 5,248[b] (753–24,528) | <0.001 |
| D-erythro-sphingosine | 7,341[a] (4,159–39,450) | 8,433[a] (2,795–14,346) | 10,575[a] (1,003–22,798) | 3,460[a] (620–20,251) | 2,778[b] (869–6,781) | 21,981[a] (438–49,033) | <0.001 |
| Isopentadecanoic acid | 16,537[b] (8,419–47,233) | 19,852[a] (6,576–44,308) | 32,099[b] (1,673–766,411) | 5,119[b] (1,627–17,766) | 25,695[b] (7,307–84,652) | 65,848[c] (21,866–707,919) | <0.001 |
| **Polyamines** | | | | | | | |
| Putrescine | 325,567[a] (97,835–1,380,413) | 129,926[a] (44,868–363,883) | 38,927[b] (27,370–102,805) | 83,581[a] (28,669–207,557) | 60,980[a] (11,066–212,790) | 201,011[a] (6,192–485,041) | 0.01 |
| **Antioxidants / antimicrobials** | | | | | | | |
| 3,4-dihydroxyhydrocinnamic acid | 20,308[a] (11,158–79,529) | 19,951[a] (9,793–36,089) | 6,608[b] (3,079–14,086) | 7,699[b] (4,398–16,818) | 28,082[a] (1,849–40,617) | 26,438[a] (1,485–61,532) | <0.001 |
| 3,4-dihydroxycinnaminic acid | 4,356[a] (1,269–10,007) | 6,229[a] (1,990–25,507) | 2,543[a] (1,178–16,989) | 2,955[a] (1,481–18,549) | 1,468[b] (281–6,545) | 760[b] (161–4,120) | <0.001 |
| 3,4-dihydroxybenzoic acid | 4,141[a] (1,918–12,984) | 4,062[a] (2,054–13,140) | 2,114[b] (1,533–3,328) | 2,489[b] (1,694–4,578) | 13,395[c] (2,783–26,592) | 6,790[c] (518–26,569) | <0.001 |
| 4-hydroxybenzoate | 6,538[a] (3,972–27,743) | 11,523[a] (4,231–17,030) | 1,035[b] (665–1,411) | 1,201[b] (619–1,940) | 16,773[c] (8,232–23,719) | 16,643[c] (420–49,060) | <0.001 |

**Notes.**

[*]*P*-values adjusted based on the Benjamini and Hochberg False discovery rate (fdr).

Profiles that do not share a common superscript letter differed significantly (fdr-adjusted *P* < 0.05) based on post-hoc analysis.

**Table 6  Metabolic pathways that differed significantly over time.** Pathways that differed significantly over time by group based on fecal metabolite profiles in feces collected at baseline (days 5–7), at the conclusion of antibiotic administration (days 26–28), and after a 603 day washout (days 631–633) from 16 healthy cats, eight per group, + that received 150 mg clindamycin with either a placebo or synbiotic PO once daily for 21 days. +Feces from one cat (synbiotic group) unavailable at the days 631–633 timepoint.

| | | Total Cmpds | Hits | fdr-adjusted *P*-value | Pathway impact value |
|---|---|---|---|---|---|
| 1 | Linoleic acid metabolism | 15 | 1 | 0.018 | 0.66 |
| 2[a] | Alanine, aspartate and glutamate metabolism | 24 | 9 | 1.54E–10 | 0.59 |
| 3[b] | Galactose metabolism | 41 | 16 | 2.00E–08 | 0.58 |
| 4[a] | Glycine, serine and threonine metabolism | 48 | 9 | 4.69E–09 | 0.54 |
| 5[a] | Arginine and proline metabolism | 77 | 16 | 1.54E–10 | 0.48 |
| 6[a] | Taurine and hypotaurine metabolism | 20 | 4 | 1.37E–08 | 0.38 |
| 7[b] | Starch and sucrose metabolism | 50 | 8 | 2.87E–08 | 0.36 |
| 8[a] | beta-Alanine metabolism | 28 | 6 | 7.25E–09 | 0.34 |
| 9 | Pyruvate metabolism | 32 | 2 | 3.82E–06 | 0.32 |
| 10[a] | Phenylalanine metabolism | 45 | 13 | 1.18E–08 | 0.27 |
| 11[c] | Pantothenate and CoA biosynthesis | 27 | 6 | 2.77E–09 | 0.25 |
| 12[a] | Cysteine and methionine metabolism | 56 | 8 | 4.43E–09 | 0.23 |
| 13 | Nicotinate and nicotinamide metabolism | 44 | 4 | 1.43E–08 | 0.23 |
| 14[a] | Lysine degradation | 47 | 4 | 1.50E–02 | 0.23 |
| 15[d] | Pyrimidine metabolism | 60 | 9 | 6.20E–08 | 0.22 |
| 16 | Arachidonic acid metabolism | 62 | 1 | 1.01E–04 | 0.22 |
| 17 | Glycerolipid metabolism | 32 | 5 | 2.80E–08 | 0.22 |
| 18 | Histidine metabolism | 44 | 3 | 1.43E–08 | 0.21 |
| 19[ad] | Amino sugar and nucleotide sugar metabolism | 88 | 9 | 1.37E–08 | 0.21 |
| 20[a] | Tyrosine metabolism | 76 | 10 | 1.96E–09 | 0.20 |
| 21 | Aminoacyl-tRNA biosynthesis | 75 | 18 | 2.84E–10 | 0.17 |
| 22[c] | Butanoate metabolism | 40 | 7 | 9.61E–11 | 0.17 |
| 23[a] | Tryptophan metabolism | 79 | 2 | 2.77E–09 | 0.16 |
| 24[a] | D-Glutamine and D-glutamate metabolism | 11 | 2 | 3.82E–06 | 0.14 |
| 25 | Inositol phosphate metabolism | 39 | 1 | 2.60E–03 | 0.14 |
| 26 | Citrate cycle (TCA cycle) | 20 | 3 | 9.75E–06 | 0.12 |
| 27 | Sulfur metabolism | 18 | 4 | 1.59E–08 | 0.11 |
| 28[b] | Pentose phosphate pathway | 32 | 5 | 8.38E–12 | 0.11 |
| 29[d] | Purine metabolism | 92 | 12 | 1.61E–10 | 0.11 |
| 30[a] | Phenylalanine, tyrosine and tryptophan biosynthesis | 27 | 5 | 4.90E–09 | 0.11 |
| 31[a] | Lysine biosynthesis | 32 | 2 | 3.07E–03 | 0.10 |
| 32 | Glycolysis or Gluconeogenesis | 31 | 4 | 2.12E–06 | 0.10 |
| 33 | Sphingolipid metabolism | 25 | 3 | 5.39E–04 | 0.09 |
| 34[b] | Pentose and glucuronate interconversions | 53 | 7 | 7.42E–09 | 0.09 |
| 35 | Glycerophospholipid metabolism | 39 | 2 | 3.38E–07 | 0.09 |
| 36[c] | Propanoate metabolism | 35 | 6 | 2.23E–06 | 0.09 |
| 37[a] | Valine, leucine and isoleucine biosynthesis | 27 | 6 | 9.82E–07 | 0.09 |
| 38[b] | Fructose and mannose metabolism | 48 | 3 | 4.94E–04 | 0.07 |
| 39 | Nitrogen metabolism | 39 | 10 | 2.77E–09 | 0.07 |

**Table 6** (*continued*)

|  |  | Total Cmpds | Hits | fdr-adjusted *P*-value | Pathway impact value |
|---|---|---|---|---|---|
| 40[a] | Valine, leucine and isoleucine degradation | 40 | 4 | 1.84E–04 | 0.06 |
| 41 | Ubiquinone and other terpenoid-quinone biosynthesis | 36 | 4 | 2.84E–10 | 0.05 |
| 42 | Glyoxylate and dicarboxylate metabolism | 50 | 4 | 1.18E–06 | 0.04 |
| 43 | Caffeine metabolism | 21 | 1 | 1.84E–04 | 0.03 |
| 44 | Glutathione metabolism | 38 | 5 | 3.28E–10 | 0.03 |
| 45 | Ascorbate and aldarate metabolism | 45 | 3 | 2.59E–08 | 0.02 |
| 46 | Vitamin B6 metabolism | 32 | 1 | 1.47E–05 | 0.02 |
| 47 | Methane metabolism | 34 | 2 | 1.30E–05 | 0.02 |

**Notes.**

Total Cmpds, total compounds in pathway.

[a]amino acid metabolism.

[b]sugar metabolism.

[c]short-chain fatty acid metabolism.

[d]nucleotide metabolism.

to increase the likelihood of detecting a difference between groups because the incidence of AAGS due to clindamycin was unknown. Ironically, this might have obscured positive results due to the severity of AAGS induced. The synbiotic dose chosen for the study was based on a prior report describing clinical improvement in cats with chronic enteropathy treated with the same synbiotic (*Hart et al., 2012*), but use of a higher dose might be required to prevent AAGS. We elected to administer the synbiotic concomitantly with clindamycin because compliance with medication regimens is inversely correlated with the number or frequency of treatments (*Adams et al., 2005*; *van den Bemt et al., 2016*), primarily due to forgetfulness and/or logistical difficulties of balancing treatment schedules against other daily responsibilities (*Col, Fanale & Kronholm, 1990*). Although macrolide resistance occurs in a relatively high percentage of *Enterococcus faecium* and *Lactobacillus* strains, as well as a *Bifidobacterium animalis* probiotic (*European Food Safety Authority, 2012*; *Ouwehand et al., 2016*; *Strompfová, Lauková & Ouwehand, 2004*), antibiotic sensitivity of the bacterial strains in the study drug were not determined so neutralization due to concomitant antibiotic administration cannot be ruled out.

To further assess the potential efficacy of synbiotics in preventing AAGS in cats, our group has since performed a follow-up study (*Stokes, Price & Whittemore, 2017*) using a crossover design with a six-week washout period, a more clinically relevant clindamycin dose (mean 18 mg/kg/d), a higher synbiotic dose (10 billion *cfu*/d), and delaying administration of the synbiotic for 1 h after antibiotic administration to limit potential synbiotic neutralization. Cats receiving the synbiotic were significantly more likely to complete the initial treatment course, vomited less, and had higher food intake than cats receiving the placebo. Interesting, strong period effects were noted—suggesting ameliorative effects of synbiotics on AAGS may persist for at least 6 weeks after administration. Because the antibiotic and synbiotic doses, as well as timing of synbiotic administration, were all altered in that study, it is not possible to ascertain which factor had the greatest impact on clinical signs.

In addition to causing AAGS, clindamycin administration was associated with marked changes in the fecal microbiome and metabolome in this study. Alterations in alpha diversity (Shannon index, Goods coverage, Chao1 metric), beta diversity, and the dysbiosis
index returned to baseline levels by the end of the washout period. The return of diversity at a global level was not due to return to baseline abundances for individual OTUs, however. Instead, changes in relative abundances of many individual OTUs persisted through the terminal sampling timepoint (>600 days after discontinuation of antibiotic therapy) in both treatment groups. Changes included reductions in the abundances of *Actinobacteria*, including *Bifidobacterium*, *Collinsella,* and *Slackia*; *Bacteroidetes*, including *Bacteroides*; many members of the *Clostridium* clusters IV and XIV, including *Lachnospiraceae* (*Blautia*, *Coprococcus*, and *Roseburia*), *Ruminococcaceae* (*Faecalibacterium* and *Ruminococcus*), and *Erysipelotrichaceae* (*Bulleidia* and [*Eubacterium* ]); and *Veillonellaceae*, including *Megamonas*. Marked increases in abundance were identified for *Clostridiaceae*, including *Clostridium*, and *Proteobacteria*, including *Aeromonadales* but primarily due to increases in *Enterobacteriaceae*. Alterations were generally consistent with those found in cats (*Ramadan et al., 2014*; *Suchodolski et al., 2015*) and people (*Schubert et al., 2014*) with dysbiosis due to naturally-occurring diarrhea. Interestingly, decreased relative abundances of *Lachnospiraceae* and *Ruminococcaceae* also have been found in cats infected with *Giardia* and feline immunodeficiency virus (*Šlapeta et al., 2015*; *Weese et al., 2015*).

Antibiotic-induced dysbiosis and gastrointestinal signs have been associated with alterations in a variety of metabolite profiles. Consistent with prior work in people (*Gustafsson et al., 1998*; *McFarland, 2008*; *Woodmansey et al., 2004*), we found marked derangements in SCFA profiles and pathways during antibiotic administration, some of which persisted at days 631–633. No difference was found in SCFA profiles between treatment groups. In addition to being an important energy source for colonocytes, SCFAs have anti-inflammatory and immunomodulatory effects, regulate intestinal motility, and maintain intestinal barrier function (*Quigley, 2012*; *Suchodolski, 2016a*). Persistent alterations in SCFA profiles could perpetuate ongoing dysbiosis after resolution of the initial insult.

Decreased proportions of secondary bile acids, important for maintaining eubiosis and modulating the immune system, have been found in people and dogs with gastrointestinal disease (*Duboc et al., 2013*; *Honneffer et al., 2015*). Similarly, we identified a marked reduction in secondary versus primary bile acids during antibiotic administration, followed by an increase at days 631–633 above baseline. It is possible, though speculative, that the inversion in the ratio of primary to secondary bile acids on days 631–633 reflects a compensatory response to persistent, occult dysbiosis and inflammation. Temporal alterations in bile acid biosynthesis were also noted on pathway analysis but did not achieve significance (fdr adjusted $P = 0.06$).

Indole profiles, the product of tryptophan degradation by commensal *E. coli*, have previously been found to be decreased in people and animals with gastrointestinal disease (*Suchodolski, 2016a*). In this study, indole profiles decreased significantly during antibiotic administration in both treatment groups. Indole profiles failed to return to baseline values by days 631–633 for cats in the placebo group, while they were mixed for cats in the synbiotic group. Direct effects of indole within the gut lumen include decreased production of adherence and virulence factors, motility, biofilm formation, and adherence of pathogenic *E. coli* (*Bansal et al., 2007*), while effects on the intestine include strengthening of intestinal tight junctions, increased production of mucin to shield the epithelium from bacterial

adherence, and modulation of inflammation (*Bansal et al., 2010*). Persistent alterations in indole profiles in cats not administered a synbiotic with antibiotics could contribute to a pro-inflammatory state, as well as chronically increased intestinal permeability and risk of bacterial colonization and translocation.

Significant temporal changes also were found in profiles of D-erythro-sphingosine, isopentadecanoic acid, and cellobiose. D-erythro-sphingosine is a sphingolipid, while isopentadecanoic acid and cellobiose are components of sphingolipids and sphingophospholipids (*Chao, Khan & Hannun, 1992*; *Minamino et al., 2003*; *Vesper et al., 1999*). Sphingolipids and sphingophospholipids have antineoplastic properties, stimulate the innate immune system, and increase bacterial clearance (*Chao, Khan & Hannun, 1992*; *Fujiwara et al., 2013*; *Minamino et al., 2003*). Persistent reductions in putrescine profiles in cats in the placebo, but not the synbiotic, group after antibiotic exposure also were noted. Putrescine, along with spermine and spermidine, are polyamines. Polyamine supplementation increases resistance to oxidative stress in mice, while depletion is associated with development of pancreatitis (*Minois, Carmona-Gutierrez & Madeo, 2011*). Further study will be necessary to determine the clinical relevance of differing sphingolipid and putrescine profiles between treatment groups over time.

Finally, significant associations were found in profiles for cinnaminic acid and benzoic acid compounds. Compounds in both treatment groups had lower results at days 26–28 than baseline. Benzoic acid profiles were higher at days 631–633 than at baseline, while cinnaminic profiles did not return to baseline levels. Both cinnaminic and benzoic acids have been found to have antimicrobial effects, and cinnaminic acids have additional anti-inflammatory properties (*Alam et al., 2016*; *Manuja et al., 2013*). The persistent decrease in cinnaminic acid profiles after antibiotic administration is particularly intriguing given work in research models suggesting potential benefits of cinnaminic acids in management of obesity and diabetes mellitus (*Alam et al., 2016*). Repeated exposure to antibiotics has been associated with increased risks of childhood obesity and diabetes mellitus (*Mikkelsen et al., 2016*; *Scott et al., 2016*). It is possible, though certainly speculative at this juncture, that associations between antibiotic exposure and increased risk of obesity and diabetes are at least partially due to cumulative derangements in cinnaminic acid profiles.

One factor that could impact clinical application of these results is the use of healthy research cats. Creation of AAGS in healthy privately-owned animals for antibiotic-related investigations poses concerns regarding compliance with the study protocol, while evaluations in clinically ill animals can be confounded by heterogeneity in underlying disease, the antibiotic required for the primary disease, prior antibiotic exposure, and husbandry factors. The predominant phylum identified at baseline in this study, *Actinobacteria*, conflicted with that found in recent reports in privately-owned healthy cats, *Firmicutes* (*Bell et al., 2014*; *Suchodolski et al., 2015*; *Weese et al., 2015*), though they agree with findings of other studies, particularly those using research colony cats (*Abecia et al., 2010*; *Desai et al., 2009*; *Jia et al., 2011a*; *Jia et al., 2011b*). Although *Firmicutes* was the predominant phyla in the first 3 studies (*Bell et al., 2014*; *Suchodolski et al., 2015*; *Weese et al., 2015*), relative abundance ranged from 50–80% with similar variability in *Actinobacteria*

(0.11–9.9%), *Bacteroidetes* (0.15–33%), *Fusobacteria* (0–1.15%) and *Proteobacteria* (0–30%). Differences among studies also could reflect geographical variation, heterogeneity in diet, or variable exposure to other treatments known to impact the microbiome. Although cats in this study had not previously received antibiotics, probiotics or proton-pump inhibitors, data on potential exposure to them were not included in the other reports.

Because there was not a control group that received neither antibiotics nor probiotics, it is not possible to definitively ascribe long-term changes in the fecal microbiome and metabolome to antibiotic administration. Although the fecal microbiome of cats evolves with age, predominant bacterial groups and relative abundances appear relatively stable in colony cats based on evaluation of cats from one to nine months of age and eight to 14 years from the same colony (*Jia et al., 2011a*; *Jia et al., 2011b*). Further, we did not identify an association between the age of cats and the composition of the microbiome at baseline. As such, exposure to antibiotics is the most likely cause for the marked changes in fecal microbiome and metabolomic profiles over time. Interestingly, we also found no association between age and AAGS, in contrast to work in people (*McFarland, 2008*).

## CONCLUSIONS

High-dose clindamycin therapy induces AAGS in 100% of cats, and severity of signs is often treatment-limiting. The incidence of vomiting was lower in cats concurrently administered a synbiotic, although the difference was not statistically significant. Antibiotic-induced dysbiosis and alterations in fecal metabolite profiles mirror those found in people. Changes in fecal metabolomic profiles included derangements in SCFA, bile acid, indole, sphingolipid, polyamine, benzoic acid, and cinnaminic acid metabolism. Changes in relative abundances of many OTUs and metabolite profiles persisted for >600 days after treatment discontinuation. Significant differences between synbiotic and placebo groups were noted for metabolites that affect immunomodulation, intestinal permeability and barrier function, colonization resistance, and oxidative stress. Further investigation is warranted to determine whether synbiotics blunt AAGS or the risk of antibiotic-associated metabolic derangements, such as obesity, in cats undergoing repeated antibiotic therapy.

### Funding

This project was supported by the University of Tennessee, College of Veterinary Medicine Companion Animal Fund; University of Tennessee, College of Veterinary Medicine Center of Excellence in Livestock Diseases and Human Health Summer Research Program; and Nutramax Laboratories Veterinary Sciences, Inc., Lancaster, SC. The funders had no involvement in the design or performance of the study, writing of the manuscript, or the decision to submit the manuscript for publication. The funders had no role in study design, data collection and analysis, decision to publish, or preparation of the manuscript.

### Grant Disclosures

The following grant information was disclosed by the authors:

University of Tennessee, College of Veterinary Medicine Companion Animal Fund.
University of Tennessee, College of Veterinary Medicine Center of Excellence in Livestock Diseases and Human Health Summer Research Program.
Nutramax Laboratories Veterinary Sciences, Inc., Lancaster, SC.

## Competing Interests

Two of the investigators (JCW, JSS) declare past receipt of honorariums from Nutramax Laboratories for public speaking and educational materials.

## Author Contributions

- Jacqueline C. Whittemore conceived and designed the experiments, performed the experiments, analyzed the data, prepared figures and/or tables, authored or reviewed drafts of the paper, approved the final draft.
- Jennifer E. Stokes conceived and designed the experiments, performed the experiments, analyzed the data, authored or reviewed drafts of the paper, approved the final draft.
- Nicole L. Laia performed the experiments, authored or reviewed drafts of the paper, approved the final draft.
- Joshua M. Price analyzed the data, contributed reagents/materials/analysis tools, prepared figures and/or tables, authored or reviewed drafts of the paper, approved the final draft.
- Jan S. Suchodolski performed the experiments, analyzed the data, contributed reagents/materials/analysis tools, prepared figures and/or tables, authored or reviewed drafts of the paper, approved the final draft.

## Animal Ethics

The following information was supplied relating to ethical approvals (i.e., approving body and any reference numbers):

This study was approved by the Institutional Animal Care and Use Committee of the University of Tennessee, Knoxville (protocol number 2169) and performed in compliance with ''The Guide for the Care and Use of Laboratory Animals'' in laboratory animal facilities that are AAALAC certified and exceed NIH standards of care.

## Data Availability

The microbiome dataset generated and analyzed during the current study is available in the SRA repository, accession number SRP08225 (https://www.ncbi.nlm.nih.gov/sra). The metabolomics data is available at the website of the NIH Common Fund's Metabolomics Data Repository and Coordinating Center (supported by NIH grant, U01-DK097430), the Metabolomics Workbench (http://www.metabolomicsworkbench.org), where it has been assigned study number ST000979.

Clinical and metabolomic data have been supplied as Supplementary Files.

## Supplemental Information

Supplemental information for this article can be found online at http://dx.doi.org/10.7717/peerj.5130#supplemental-information.

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
