# Peer review of "Short and long-term effects of a synbiotic on clinical signs, the fecal microbiome, and metabolomic profiles in healthy research cats receiving clindamycin: a randomized, controlled trial"

_PeerJ, doi:10.7717/peerj.5130_

## Round 0.1 · original submission · Major Revisions

First of all, I have to apologize for the extensive delay in reaching a decision. It turned out to be extremely difficult to secure more than 1 expert reviewer, and a second reviewer could only now be found.
Both reviewers pointed out a number of major reservations especially regarding experimental set-up, focus of the study and the fact that a more advanced experiment has already been published by now.

In case you feel that you can address these and other major comments sufficiently, I would be happy to evaluate a revised manuscript.

·

Basic reporting

This paper is clearly written in professional English throughout. Literature references and field background certainly are sufficient. Article structure, figures, tables and raw data are clear. There is, however, as the authors admit themselves in the discussion, a problem with the relevance of the results to the hypothesis (a synbiotic will affect food intake, vomiting, fecal scores, fecal matabolome and fecal microbiome during a treatment with clindamycin): insufficient power due to low number of animals. The discussion is very lengthy, with many repetitions.

Experimental design

this is original primary research within aims and scope of the journal. The research question is well defined, relevant and meaningful. The knowledge gap is, however, essentially limited to cats. The local ethincal committee has evaluated the ethical standard. One may question whether this investigation was carried out rigorously. indeed, Indeed, the time lapse between defecation and collection and safe storage of fecal samples is not indicated and may account for the lack of clear and unambiguous conclusions with respect to the effect of the synbiotic. Moreover, some indication is given as to the prebiotic strains included, but there is no information regarding the prebiotic included in the synbiotic. It looks as if the synbiotic contained glutamate to improve palatability, since glutamate was the most important change in the metabolome associated with treatment. there are some problems with the experimental set up and the methods used. there is no control group without antibiotic treatment. The number of animals is insufficient, resulting in underpowering.

Validity of the findings

A follow-up study with a better experimental set up by the same group has already been published (Stokes et al., 2017). Data are insufficiently robust. Sensitivity of the probiotic strains to the antibiotic was not tested. The discussion about Clostridium difficile is irrelevant in cats.

Additional comments

The effects on the microbiome could be partly explained by difference in time lapse between voiding of the faeces and safe storage.

Reviewer 2 ·

Basic reporting

1) Introduction and rationale: Although the rationale can be assumed from the abstract and introduction, the authors do not present it in a linear, focused manner. The authors start out their intro by providing the rationale that antibiotic-associated GI signs (AAGS) is prevalent, and that this is a reason for non-compliance, and immediately bring up the use of probiotics without providing any kind of data that supports the efficacy of probiotics. Example, (lines 58-59): What does “the efficacy of probiotics in preventing AAGS does not appear to hinge on the antibiotic used” mean? They use this as a rationale to support the hypothesis that “dysbiosis” is a contributor to AAGS by altering the metabolome, which is not really a logical line to that rationale. This is all in the first paragraph. The next paragraph discusses antibiotic resistance following antibiotic exposure in detail, which is not the context of the manuscript at hand. Although the next couple paragraphs link this concept back to compliance, it is not immediately clear what the relevance of doing metabolomics in this study is, to cats or humans. I would suggest re-focusing the introduction to provide a clear line of thinking that lays out the rationale for doing metabolomics on antibiotic-treated cats, and how this relates to solving issues with AAGS. What is the purpose of doing metabolomics if you can just compare the clinical results of synbiotic/antibiotic combinations? I would also add some of these concepts and rationale, particularly in relation to synbiotic use, in the ‘Background’ section of the Abstract—as it currently stands, the reader is left wondering why synbiotics were used, and how this relates to metabolomics.

Experimental design

1) Some of the methods need clarification so that these results can be compared to work of others. The commercial kit used to extract the DNA needs to be directly listed in the manuscript and not just cited. The synbiotic used needs to be listed directly, not just referenced (what substance was used?). What is MR DNA? What similarity cutoff was used to cluster the OTUs? From the way the methods are written, it is unclear if sequences were directly binned taxonomically (which would be classified using a database, and does not require OTU clustering) or if the sequences were actually clustered into OTUs using distance, then classified to a database (using a representative sequence from that OTU). Also, where were the raw sequences deposited? What database was used to identify the untargeted metabolomics profiles? Some detail should be added here. Finally, even though the general experimental timeline is given, there is no mention of how many samples were collected during the defined time periods in methods or throughout the manuscript. One suggestion is to add a “figure 1” that illustrates the study design and indicates sampling. More importantly, what “synbiotic” was used for the study? The authors never discuss relevancy of the synbiotic, and which microbes are capable of utilizing this product, or how it is purported to work.

2) Results: Some of the described data needs some clarification and more rigorous analysis.
--According to Table 1, one of the largest differences in the clinical scores of cats with or without synbiotic treatment at baseline is vomiting in synbiotic-treated cats,which then decreases by week 3, at which point placebo-treated cats had less. How exactly did the authors determine that “there was no difference between group”? In looking at the data, differences between the groups at each time point seem significantly different (example, 0+/-0 compared to 1.8+/-5.1 at baseline).
--What is the point of Figure 1? Although it does rationalize subsampling to a level of 35,000 sequences per sample, if the point is to compare differences between the groups, there are better ways of making this point, such as a diversity metric, which the authors reference in the text (line 234) but do not actually show in Figure 1. Instead of having a separate table and a rarefaction plot, Figure 1 could show Shannon diversity, goods coverage, or chao1 (or all) to make this point more efficiently.
--(line 238): The authors state that beta diversity was significantly altered, but then state that there was no difference between groups? Since the data is grouped both by a) timepoints and b) synbiotic vs. placebo groups throughout the manuscript, the authors should specify which groups (a or b) are being referenced to when referencing “groups”.
--(line 238 and Table 2): What is a “dysbiosis index”? In addition to citing this, the authors should explain what this is, and what it means. What are the superscript a’s and b’s in Table 2?
--Metabolomics results: How was change over time in the assayed compounds done (line 250)? The statistical tests/methods should be called out in the text. While organizing the samples by group makes the changes in metabolites easy to visualize, was there any unsupervised clustering that revealed shared patterns between the groups (time periods compared to each other, or between the treatment groups during antibiotics)? Similarly, the information conveyed by both Figure 4 (heatmap) and Figure 5 (PCoA from metabolites) basically convey the same information.
--The authors call out 1 metabolite out of many differences (N-acetylglutamate) that differs between placebo and synbiotic groups during antibiotics. While this molecule may indeed be relevant to the biology of the treatment, was this the only one? In the current way that the analysis and text is organized, it is difficult to distinguish which “groups” are being compared to each other (particularly without referencing what type of statistical method was used), and what cutoffs were employed to identify the specific metabolites called out. Part of this problem is using group-wise vs. longitudinal analyses interchangeably. This is a rich dataset, and some more attention could improve the analysis and interpretation of the data. One suggestion could include actually graphing significantly altered metabolites over time per cat, colored by treatment group (synbiotic vs. placebo), over time. This would demonstrate significant changes in context with the study design within the different groups, as well as be appropriate for longitudinal data visualization.
--Related to the above point, the authors also call out specific metabolites (cholesterol, CDCA) in the text as being unchanged. What is the relevance of calling out these particular metabolites? Another way to demonstrate context to the rationale would be to mention why these are metabolites of interest.

Validity of the findings

Discussion and conclusions:
--(line 298): Does the data suggest that “stimulation of the motilin receptor had minimal impact on the incidence of vomiting”? What is the motilin receptor, and how was it exactly stimulated in this study?
--The authors state that abx administration was associated with marked changes in the fecal microbiome and metabolome that persist at followup (line 302; 330-333), but figures 1 and 2 would argue otherwise. The authors also state that resolution was found at followup later on (lines 344-359). Any changes observed in the metabolome during followup sampling suggest that the metabolomics profiles are also markedly differnet from baseline samples, not just during antibiotics, suggesting a major shift over time in general independent of treatments.
--(line 307): The authors do not define a “dysbiotic state”—what would a normal cat gut microbiota community look like? What does a dysbiotic one look like? How does the authors’ data compare to other cited data in terms of community members identified, healthy or otherwise? These types of broad statements would be greatly strengthened by having some detailed definitions or comparisons.
--(line 310-313): What does the statement “AAGS can be delayed in onset by as much as 8 weeks with recurrence of antibiotic-associated C. diff infection in up to 50% of individuals” mean? C. diff infection is a very specific type of antibiotic-associated diarrheal case, and if there is a recurrence, you are not experiencing AAGS, you are experiencing recurrent CDI, which is an entirely different disease definition.
--(line 317-320): This section is unclear. Why would “efficacy of probiotics in preventing AAGS suggest that disruptive effects of antibiotics on the microbiome are a primary contributor to AAGS”? The data on probiotic efficacy against anything are complicated and in some cases controversial. While the prospect of probiotics is of interest, the authors should be careful and detailed in their statements concerning probiotics. On a similar note, the authors should be careful in over-interpreting published work in their discussion points about AAGS, “dysbiotic” microbiota, and pathogen colonization/infection (lines 324-333).
--All of the discussion in lines 344-596 (note: this is 8 pages) is either out of the scope of the current manuscript, listings of different types of microbiota profiles found in various animals, or actual data that was never referenced in the results (lines 334-343). The discussion topics range from data modeling human CDI (a very different clinical definition), to detailed discussion about particular metabolite groups, to antibiotic resistance found in probiotics. The authors need to put their data into the context of their rationale, original aims, and the relevant work, preferably in 2, 3 pages maximum.
--(lines 599-609): One of the concluding statements by the authors is that “antibiotic-induced changes in bacterial diversity and taxonomic bundances closely mirror those found in people”, yet this is not what is obvious by the discussion points. It is also not clear what the actual conclusions are, other than re-stating the results. Why exactly does this warrant “whether persistent antibiotic-induced dysbiosis is a risk factor for AAGS secondary to repeated antibiotic exposure”? There are many themes present throughout the manuscript (preventing AAGS with synbiotics, using cats as models of human interventions, identifying metabolic biomarkers from metabolomics data that might indicate microbiota-induced GI infections). My suggestion would be to focus on one or two themes/questions out there (provide rationale initially), assess what can be concluded from the study at hand, and how this changes or advances our ability to treat the stated problem. This would strengthen the manuscript as a whole.

Additional comments

General Comments
- In the Abstract results section, there needs to be some context. For instance, what does “normalization by days 631-633” mean? Day relative to what—antibiotics? Synbiotic treatment? Baseline?
- The manuscript uses a combination of citation with a superscript letter, and full citations (author and year) throughout the paper. In most cases, footnotes should be citations, or the facility can just be listed as is. Whatever the correct format is for PeerJ, consistency is necessary.
- (in methods and lines 213-214) The authors can remove mention of cats that were removed from study, and just concentrate on the screening for the 20 cats actually included, for simplicity.
- Pertaining to the ethics statement, there should be a statement about welfare and enrichment of the cats in the facility.
- (lines 219-220): should specify that none of the cats had abnormal clinical scores at RECOVERY time (if this is the case).
- Statistical tests applied to data should be listed in their respective text and legends. This is important.

---

## Round 0.2 · Minor Revisions

I agree with both reviewers that the paper has strongly improved. One of the reviewers has raised a few remaining issues related to lacking info on qPCR in the methods section, as well as the need for a better focus of the discussion.

·

Basic reporting

The authors have responded to the remarks made by the referees is a careful and correct way. I have no further remarks (except for one typo on line 129).

Experimental design

no comment

Validity of the findings

no comment

Reviewer 2 ·

Basic reporting

Results within the manuscript are improved by adding a figure demonstrating the study design and collating the reported metabolomics data into figures rather than a long table. Overall, the authors are very detailed in their reporting.

Experimental design

The methods have been largely clarified by the authors in this resubmission. In the description of the dysbiosis index, the authors state "the dysbiosis index is a quantitative PCR based assay that quantifies the abundances of 8 major bacterial groups". However, I do not see a description for quantitative PCR in the methods?

The lack of control groups (cats without antibiotics and cats receiving the synbiotic alone) made some of the study conclusions about how probiotics might change the microbiome/metabolome difficult to make, but the authors have now included a statement addressing this in the discussion.

Validity of the findings

The discussion section is still long and in need of refocusing. The authors discuss a lot of concepts in a fair amount of detail. Given the results and analysis presented in the study, I am not sure that each compound needs its own paragraph detailing all the mechanisms/studies that have been published for that compound. One would expect that many metabolites will decrease following antibiotics, so it is unclear what a decrease in a specific group of metabolites means. Having such detail for each metabolite group is highly speculative, and the message of the study is lost. This section should be condensed to address the impact the authors' results have that pertain to their original study question. The authors also have some results in this section (lines 618-623) that were not presented in the results section.

Minor comment: I am not sure what this statement means (lines 613-615): "Consistent with other reports (Schubert et al. 2014), the return of diversity at a global level was not due to return to baseline abundances for individual OTUs". Additionally, the cited Schubert et al study was not a longitudinal study comparing baseline measurements.

---

## Round 0.3 · accepted · Accept

Thanks for the further improvement of the paper.

Indeed, data presented in your manuscript are of significant interest, and I am looking forward to seeing in published.

Regarding the concerns of the reviewer with respect to too much detail on the metabolites, and a too lengthy discussion in general, you could consider adding a few subtitles in the discussion to allow the reader to read it all, or rather focus on specific issues.

#